# High resolution mapping of the tumor microenvironment using integrated single-cell, spatial and in situ analysis

Amanda Janesick[1], Robert Shelansky[1], Andrew D. Gottscho[1], Florian Wagner [1], Stephen R. Williams[1], Morgane Rouault[1], Ghezal Beliakoff[1], Carolyn A. Morrison [1], Michelli F. Oliveira[1], Jordan T. Sicherman [1], Andrew Kohlway[1], Jawad Abousoud[1], Tingsheng Yu Drennon [1], Seayar H. Mohabbat[1], 10x Development Teams* & Sarah E. B. Taylor [1] ✉

Single-cell and spatial technologies that profile gene expression across a whole tissue are revolutionizing the resolution of molecular states in clinical samples. Current commercially available technologies provide whole transcriptome single-cell, whole transcriptome spatial, or targeted in situ gene expression analysis. Here, we combine these technologies to explore tissue heterogeneity in large, FFPE human breast cancer sections. This integrative approach allowed us to explore molecular differences that exist between distinct tumor regions and to identify biomarkers involved in the progression towards invasive carcinoma. Further, we study cell neighborhoods and identify rare boundary cells that sit at the critical myoepithelial border confining the spread of malignant cells. Here, we demonstrate that each technology alone provides information about molecular signatures relevant to understanding cancer heterogeneity; however, it is the integration of these technologies that leads to deeper insights, ushering in discoveries that will progress oncology research and the development of diagnostics and therapeutics.

High-throughput methods in single-cell genomics have made it possible to cluster thousands to millions of cells from a single experiment into distinct types based on whole transcriptome gene expression and cell surface protein data, sparking ambitious collaborations to profile every cell type in the human body[1–5]. Meanwhile, advances in spatial transcriptomics have introduced unbiased gene expression analysis with spatial context for tissue sections, combining genomics, imaging, and tissue pathology[6,7]. These technologies are complementary in that single-cell methods lack spatial context, while spatial methods may require integration with single-cell data to infer detailed information about cell type composition. There has been significant progress in integrating these data types through transcript distribution prediction and cell type deconvolution[8]. However, analysis of high-resolution cell–cell and ligand–receptor interactions that comprise intercellular communication is lacking, as is the definitive assignment of transcripts to a particular cell with spatial context at high gene plexy. An ideal solution would provide high-plex, high throughput, multi-modal readouts with spatial context and subcellular resolution, without compromising tissue integrity, and be compatible with both fresh frozen (FF) and formalin-fixed paraffin embedded (FFPE) tissues. Several high plex in situ technologies have recently been commercialized to address these needs and include: CosMx (NanoString), MERSCOPE (Vizgen), Molecular Cartography (Resolve), and Xenium In Situ (10x Genomics). However, a major challenge remains in integrating these data types with whole transcriptome single cell or spatial data.

Here, we use single cell, spatial and in situ technologies on serial sections of an FFPE-preserved breast cancer block to explore the heterogeneity within the tumor (Fig. 1). We use Chromium Single Cell

[1]10x Genomics Inc., Pleasanton, CA 94566, USA. *A list of authors and their affiliations appears at the end of the paper.
✉e-mail: correspondence@10xgenomics.com; sarah.taylor@10xgenomics.com

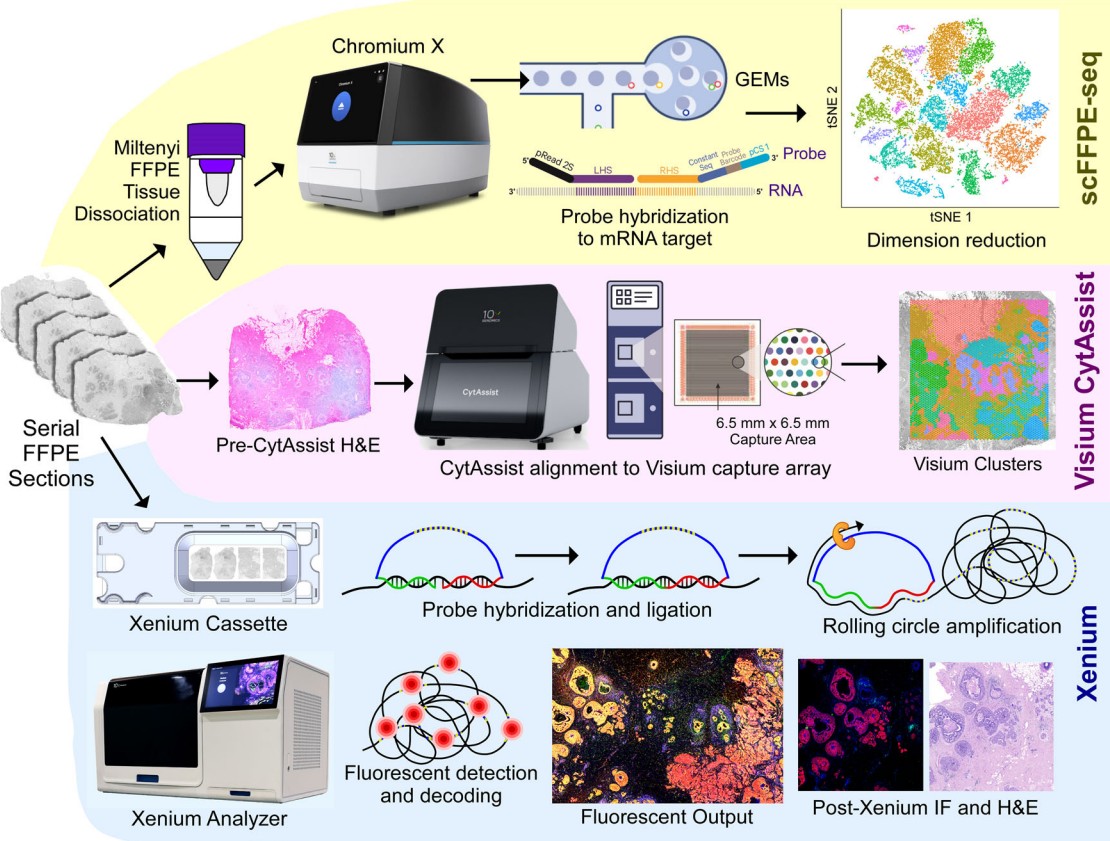

**Fig. 1 | Experimental design.** A single FFPE tissue block was analyzed with a trio of complementary technologies. Top: the Chromium Single Cell Gene Expression Flex workflow with the Miltenyi FFPE Tissue Dissociation protocol (scFFPE-seq). Middle: Visium CytAssist enabled whole transcriptome analysis with spatial context, and was readily integrated with single-cell data from serially adjacent FFPE tissue sections. Bottom: The Xenium In Situ technology uses a microscopy based readout. A 5 µm tissue section was sectioned onto a Xenium slide, followed by hybridization and ligation of specific DNA probes to target mRNA, followed by rolling circle amplification. The slide was placed in the Xenium Analyzer instrument for multiple cycles of fluorescent probe hybridization and imaging. Each gene has a unique optical signature, facilitating decoding of the target gene, from which a spatial transcriptomic map was constructed across the entire tissue section. The Xenium data could be easily registered with post-Xenium immunofluorescence (IF)/H&E images (as the workflow is non-destructive to the tissue) and integrated with scFFPE-seq and Visium data. Metrics from these experiments are contained in Supp. Table 1.

Gene Expression Flex (RNA templated ligation (RTL) technology) applied to FFPE tissues (scFFPE-seq), which unlocks vast biobanks of samples while also improving sensitivity[9]. We use Visium CytAssist to obtain whole transcriptome spatial data. Xenium provides subcellular spatial resolution, which is particularly suited for studying tumor invasion in ductal carcinoma in situ (DCIS), due to its high molecular complexity and close proximity of different cell types. Using human breast cancer cell atlas data, we selected 313 genes of interest for the targeted Xenium In Situ panel. We show how three tumor subtypes differ in their microenvironment, particularly with respect to distinct myoepithelial cell populations, and how invasive cells intrude upon a DCIS domain. We demonstrate how whole transcriptome and targeted in situ data can be integrated to provide complementary and additive biological information. Combining scFFPE-seq and Visium CytAssist enhances the annotation of cell types in the sample, which is further refined by mapping the transcripts to the Xenium data. The large imageable area provided by Xenium permits us to comprehensively explore two tumor samples and identify rare cell types. Downstream H&E staining on the same tissue section renders a useful registration of morphology and RNA, thus fostering comparison between the molecular readout from Xenium with pathologists' annotations. In the first tissue block (Sample #1), the Xenium data allows us to identify a cell type positive for the RNA of three breast cancer classifying receptors (estrogen, progesterone, and HER2) that the other technologies did not detect. We derive high resolution spatially resolved whole transcriptome information for this group of cells through integration of Visium and Xenium data, revealing differentially expressed genes associated with the triple-positive tumor region. In the second tissue block (Sample #2), we locate a small population of "boundary cells" expressing markers for both tumor and myoepithelial cells. We then identify these cells in the single cell data from Sample #1, and derive whole transcriptome information. By studying these tissues with our integrative, multi-modal approach we are able to gain a deeper understanding of the complex and diverse network of cells within the tumor microenvironment.

## Results

### Comprehensive atlasing of human breast cancer FFPE tissues with whole transcriptome single cell and spatial analysis

Breast cancer is a complex disease of multiple pathologies—each tumor subtype has unique features and significant cellular and molecular heterogeneity. To better understand tumorigenesis and the cancer ecosystem, it is necessary to dissect cellular components and molecular profiles within the spatial context of the tumor landscape. Using discovery-based technologies, we characterized a breast cancer sample with single cell and spatial whole transcriptome analysis. First, we generated Chromium scFFPE-seq data from 2 × 25 µm FFPE curls (see "Methods") of a breast cancer block (Stage II-B, ER + /PR − / HER2 +) that were adjacent to the tissue sections used for Visium and Xenium workflows. Analysis of the scFFPE-seq data yielded 17 well-

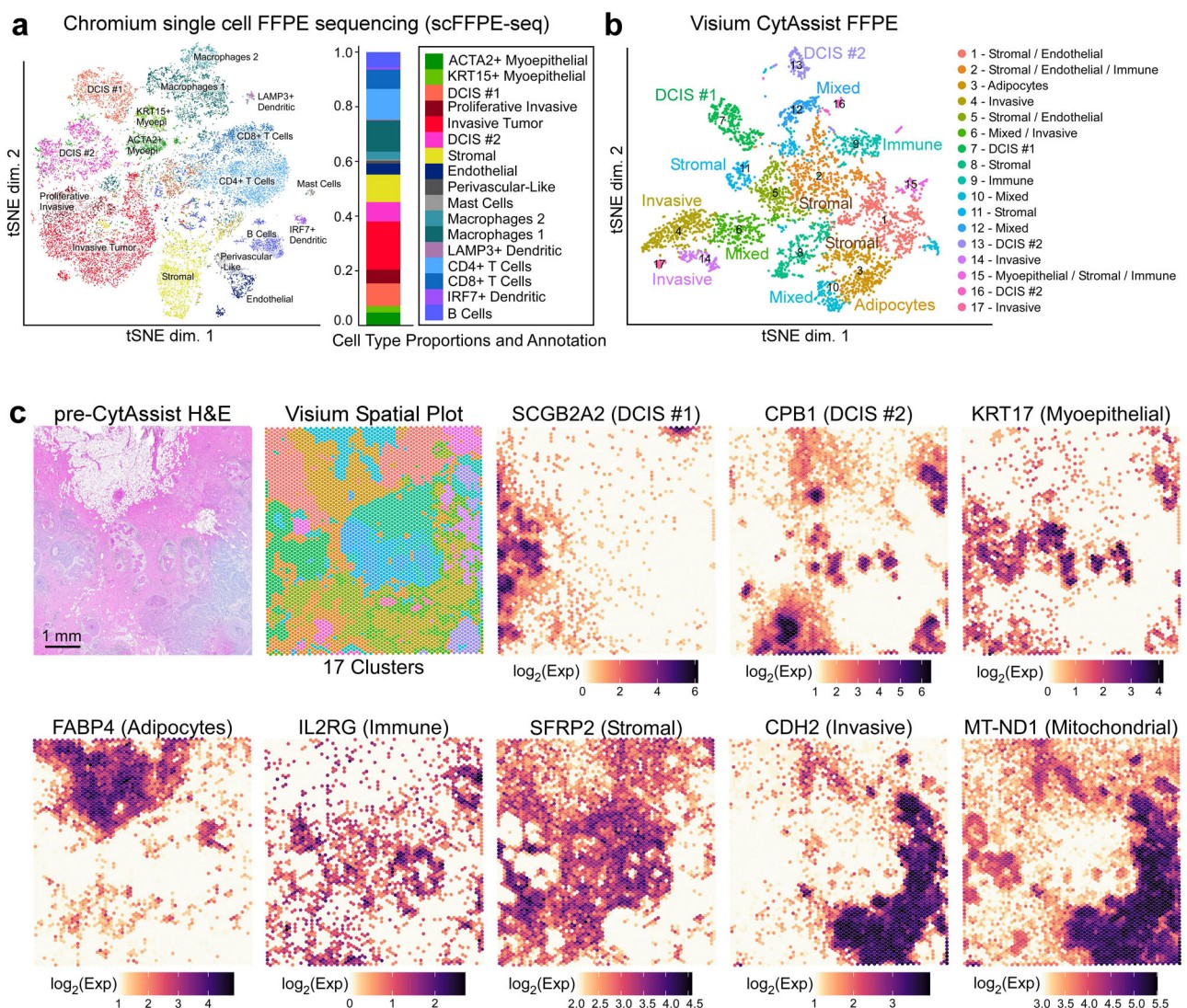

**Fig. 2 | Characterization of an FFPE-preserved breast cancer sample using whole transcriptome single cell and spatial technologies reveals complex tumor and myoepithelial heterogeneity.** A human breast cancer sample was obtained as an FFPE block (annotated by pathologist as invasive ductal carcinoma) and processed for single cell analysis and spatial transcriptomics as described in Fig. 1. **a** Dimension reduction of the scFFPE-seq data yielded a t-SNE projection with 17 unsupervised clusters. Each point represents a cell and the colors/labels show annotated cell types. Macrophages 1 cluster is marked by LYZ, IFI30, and ITGAX. Macrophages 2 cluster is marked by SELENOP, F13A1, and RNASE1. **b** t-SNE projection of Visium spots also identifies 17 clusters. Based on differential gene expression analysis, ten clusters could be unequivocally assigned to cell types, while the others were mixtures of cell types. **c** H&E staining conducted pre-CytAssist is shown for reference alongside the spatial distribution of clusters in (**b**). Scale bar = 1 mm. Cell type-specific marker genes are expressed as log$_2$(normalized UMI counts). The Visium data elucidated the spatial location of two molecularly distinct DCIS and invasive subtypes and the general locations of immune, myoepithelial, adipocytes, and stromal cells. Additionally, Visium features mitochondrial probes (e.g., MT-ND1), and their spatial distribution correlates with the invasive region of the tissue section. This experiment was performed in replicate on two serial sections, with one representative section shown here.

segregated clusters based on unsupervised clustering analysis, with a median of 1480 genes identified per cell.

Next, we generated Visium whole transcriptome data by collecting 5 μm tissue sections adjacent to those used for scFFPE-seq. Sections were H&E stained prior to imaging, followed by Visium CytAssist library preparation and sequencing. The Visium CytAssist instrument facilitates the transfer of analytes from standard glass slides to Visium slides. Both Single Cell Gene Expression Flex and Visium use the same probe set (18,536 genes targeted by 54,018 probes), allowing for easier data integration. Dimensionality reduction of the Visium data yielded 17 spatial clusters (coincidentally, the same number of clusters as the scFFPE-seq data), with a median of 5712 genes identified per spot.

We used these two discovery-based datasets and the guidance of existing human breast cancer references[4,10], to annotate the scFFPE-seq clusters (Fig. 2a) and map cell types onto the Visium data (Fig. 2b, c)

using an iterative process. Differentially expressed genes between the two macrophage groups are shown in Supp. Figure 1a. Ten Visium clusters were annotated such that they could be unequivocally assigned to cell types or disease states (Fig. 2b), while the other seven clusters had mixed cell type compositions. Visium pinpointed the spatial location of three tumor domains that were revealed as distinct clusters by scFFPE-seq, including two molecularly distinct types of ductal carcinoma in situ (DCIS), named here DCIS #1 and #2, and invasive tumor (Fig. 2c). The Visium workflow also delineated the general territory of immune and stromal cells and was able to recover transcripts from adipocytes, a delicate cell type that can float, rupture and/or stick to plastic surfaces during dissociation[11,12] (Fig. 2c).

scFFPE-seq and Visium technologies resolved cellular heterogeneity at the single-cell level and provided spatial insights, respectively. The integration of scFFPE-seq and Visium data was instrumental

to locating cell types and transcripts within the human breast cancer tissue section. However, areas where cell types coexist in close proximity could not be precisely spatially segregated within the tissue. For example, we observed substantial overlap between DCIS, myoepithelial, immune, and stromal markers, and unannotated Visium clusters representing mixtures of cell types. Given that the scFFPE-seq and Visium data collectively revealed two distinct DCIS and myoepithelial cell types, we wanted to explore the cellular neighborhoods of those regions in finer detail and higher resolution. Thus, we set out to resolve gene expression within the myoepithelial layer thinly sandwiched between the glandular epithelial cells, the basement membrane and the surrounding stroma.

## Xenium in situ data provides the deepest insights into tumor heterogeneity with spatially resolved gene expression at single-cell resolution

We next used the Xenium workflow to generate high-resolution gene expression data for a targeted panel of genes (Fig. 3). We used the Xenium Human Breast Panel (280 genes) with 33 add-on genes for a total of 313 genes, selected and curated primarily based on single cell atlas data for human breast tissues, including healthy and tumorigenic states[4,13,14] (Supp. Figure 1b). We visualized the RNA fluorescence image after one cycle of decoding, revealing the detailed structure of the tissue with high resolution (Fig. 3a). Further interrogation of the tissue allowed us to select relevant genes from the panel to identify stromal, lymphocytes, macrophages, myoepithelial, endothelial, DCIS, and invasive tumor cells (Fig. 3b). We also conducted post-Xenium H&E using standard staining protocols (Fig. 3c), allowing us to cross reference our findings with the pathologists' annotations.

Transcripts detected by Xenium were assigned to cells based on expansion of DAPI stained nuclei, expanding outwards until either 15 μm maximum distance was reached, or the boundary of another cell was reached (see "Methods"). We visualized the cell segmentation boundaries using the Xenium Explorer software (Fig. 3d), and the on-instrument pipeline outputs Xenium data in which transcripts are explicitly assigned to cells. In the section analyzed here, we observed 167,885 total cells, 36,944,521 total transcripts (Q score ≥ 20; see "Methods"), with a median of 166 transcripts per cell (Fig. 3e, f). When we downsampled the scFFPE-seq data to the 313 genes on the Xenium panel, we observed a median of 34 genes per cell for scFFPE-seq compared to a median of 62 genes per cell in the Xenium data (Fig. 3g, h). Fifty percent of total transcripts observed contribute to 27 genes (i.e., complexity measurement; Supp. Figure 2a). Observed counts of negative controls were minimal; negative control probes accounted for 0.026% of the total counts (Q ≥ 20) and decoding controls accounted for 0.01% of the total counts (Q ≥ 20) (Supp. Figure 2b).

To validate our 313-plex human breast Xenium panel, we explored the relative expression of panel genes in expected cell types. We transferred scFFPE-seq annotations to the Xenium data (supervised labeling); 86% of cells were unambiguously identified as a single cell type in the Xenium data. We filtered the scFFPE-seq data (17,696 genes; Fig. 2a) to only the 313 genes used in the Xenium human breast panel and found that the same cell type populations were identified (Fig. 3i), confirming that the Xenium human breast panel faithfully captures biological heterogeneity, although is less resolutive using only 1.8% of the whole transcriptome. The accurate assignment of transcripts to cells allows the same expected cell types to be identified from the Xenium data as from the single cell data (Fig. 3j, k, Supp. Figure 3). We mapped the localization of the cell types identified to generate a Xenium spatial plot (Fig. 3l; Supp. Figure 4), which can also be explored interactively (see Data availability). Analysis of two serial sections demonstrated the reproducibility of the technology, with the replicates having cell type proportions that were nearly identical and transcript counts that were highly correlated ($r^2 = 0.99$) (Supp. Figure 5).

Unsupervised labeling of cell types (Fig. 3j'), agnostic to the scFFPE-seq data, revealed similar annotation of cell types to Fig. 3j, however, the DCIS subtypes and proliferative tumor cells were not resolved. Furthermore, some cells were labeled as immune instead of stromal, which highlights the importance of accurate cell segmentation in tissue regions where different cell types coexist in close spatial proximity. Natural killer cells formed a subset of CD8 + T cells, and plasma cells formed a subset of B cells. Adipocytes were challenging to identify with supervised labeling since scFFPE-seq did not capture these cells (Supp. Figure 6a). Xenium, like Visium, successfully identified the location of adipocyte markers, but provided refined resolution where adipocyte transcripts skirt the edge of the cell boundary, since triglycerides fill the majority of the cell (Supp. Figure 6b–f). For this reason, adipocytes were challenging to segment, and therefore did not form a distinct cluster (Fig. 3j, j'). In conclusion, both supervised and unsupervised labeling contributed unique and complementary information to the annotation of cell types within the human breast cancer tissue.

## Xenium in situ analysis detects RNA transcripts with high sensitivity, specificity, and reproducibility

Xenium and scFFPE-seq are new technologies, and therefore, it is prudent to benchmark their sensitivity against each other and relative to existing single-cell technologies that use fresh or frozen (rather than fixed) cells. We compared these datasets to Chromium Single Cell 3' and 5' data generated from dissociated cells isolated from the same tumor. We quantified sensitivity using median gene expression such that high or low expressors would not bias our measurement. When sequencing depth was kept constant across platforms (~10,000 reads per cell), the median gene sensitivity of scFFPE-seq was higher than the existing 10x single cell platforms (Chromium 5' Gene Expression (GEX) and 3' GEX) (Supp. Fig. 7a, b). To assess the gene dropout rates, we show a Venn diagram of genes with zero counts in all cells for each Chromium technology in Supp. Fig. 7c. To benchmark Xenium and scFFPE-seq, we compared the number of transcripts per cell (Xenium) to the number of UMIs per cell (scFFPE-seq), downsampling to the number of genes on the Xenium panel. We found that Xenium is comparable in sensitivity to scFFPE-seq (Supp. Fig. 7d, and Supp. Fig. 8a).

Next, we compared Visium and Xenium data by registering the corresponding H&E images to identify the common capture area (78% of the full Visium dataset) (Supp. Fig. 8b). Since Visium probes the whole transcriptome and Xenium probes 313 genes, Visium exhibited 3.6x more total transcripts within the shared region (Supp. Fig. 8b). Visium and Xenium exhibited concordant spatial expression, exemplified by the tumor-associated epithelial marker *TACSTD2* (Supp. Fig. 8c, d). We mapped Xenium expression data onto the Visium capture area using the H&E registration information, and calculated pseudobulk counts within each Visium spot (Supp. Fig. 8e). The median gene sensitivity of Xenium across all genes on the human breast probe panel compared to Visium was 8.4x higher (Supp. Fig. 8f). To examine specificity, we compared *TACSTD2* transcript counts for Visium and Xenium and observed strong correlation ($r^2 = 0.88$) (Supp. Fig. 8g).

To validate the localization of gene expression (and specificity of the Xenium probes) we performed immunofluorescence, which is possible since the Xenium on-instrument biochemistry and decoding cycles preserve protein epitopes. HER2 (tumor) and CD20 (B cell) antibodies were detected with fluorophore-conjugated secondary antibodies and their expression was compared to their cognate RNA by overlaying the protein and RNA data together. As both images were taken from the same section, we were able to obtain a high degree of concordance and registration between the RNA and protein expression profiles (Supp. Fig. 9).

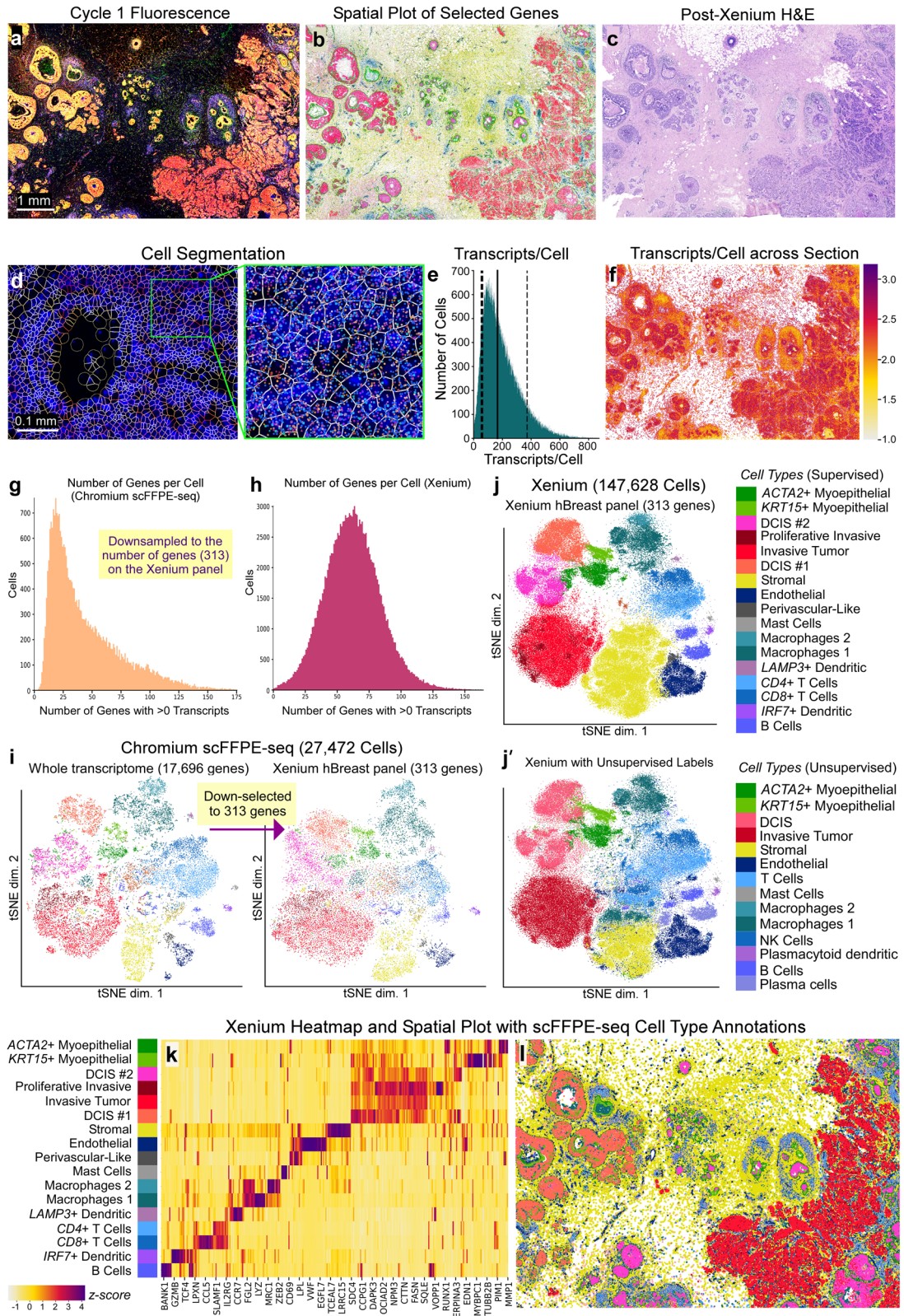

**Exploration of a breast carcinoma sample with three distinct tumor subtypes reveals heterogeneity in myoepithelial, immune, and invasive cell populations**

Ductal carcinoma in situ (DCIS) is a non-obligate precursor of invasive ductal carcinoma, which can develop into invasive disease, the treatment of which often involves surgical removal of the lesion and radiotherapy[15]. Because not all DCIS lesions progress to invasive

disease, there is great interest in understanding the molecular mechanisms underpinning invasiveness in DCIS, which are currently not well known[15,16], but could help to guide better therapeutic strategies. Our goal was to use single cell and in situ data to identify different tumor subtypes and supplement H&E imaging and pathology with molecular targets. First, we used scFFPE-seq data to map three different tumor epithelial cell subtypes and two myoepithelial subtypes to

**Fig. 3 | Xenium data provide extremely high-resolution single-cell information with spatial localization from a targeted panel of genes. a** Maximum intensity projection of RNA fluorescence signal in Cycle 1 from a 5 μm FFPE section. Fifteen of such images (unprojected, original z-stacks), one per cycle, were input into the on-instrument pipeline to decode 313 genes. Scale bar = 1 mm. **b** Selected genes representing major cell types are shown: stromal (*POSTN*, yellow), lymphocytes (*IL7R*, blue), macrophage (*ITGAX*, turquoise), myoepithelial (*ACTA2*, *KRT15*, green), endothelial (*VWF*, dark blue), DCIS (*CEACAM6*, pink), and invasive tumor (*FASN*, red). **c** H&E staining performed post-Xenium workflow, highlighting the minimal impact of the Xenium assay on tissue integrity. **d** Deep learning-based cell segmentation assigns individual transcripts to cells. Scale bar = 0.1 mm. **e** Histogram showing the distribution of transcripts per cell (Q ≥ 20). Dotted lines: 10th percentile = 61 and 90th percentile = 372 median transcripts per cell. Solid line: 50th percentile = 166 median transcripts per cell. **f** Log$_{10}$(transcripts per cell) across the entire section. **g**, **h** Bar plots showing the number of genes detected per cell for

scFFPE-seq (downsampled to the 313 genes on the Xenium panel) compared to Xenium. **i** t-SNE projection of scFFPE-seq data using all 17,696 genes (left) then down-selected to 313 genes (right). **j** t-SNE projection of Xenium cells annotated using supervised labels derived from scFFPE-seq data. Cells which were not unambiguously identified in the Xenium data (<50% of the nearest neighbors coming from one cell type) were unlabeled (~14% of cells). **j'** t-SNE projection of Xenium cells annotated using unsupervised labels, agnostic to the scFFPE-seq data. **k** Heatmap representation of the t-SNE j showing the relative expression of genes across different cell types found in the Xenium data. Scale bar is a z-score computed across cell types for each gene by subtracting the mean and dividing by the standard deviation. See Supp. Figure 3 for the corresponding scFFPE-seq heatmap. **l** Spatial plot with cell type labels transferred. The Xenium experiment was performed in replicate on two serial sections, with one representative section shown here. The scFFPE-seq data is *N* = 1 due to inherent limitations in using a single block for multiple technologies (see "Methods").

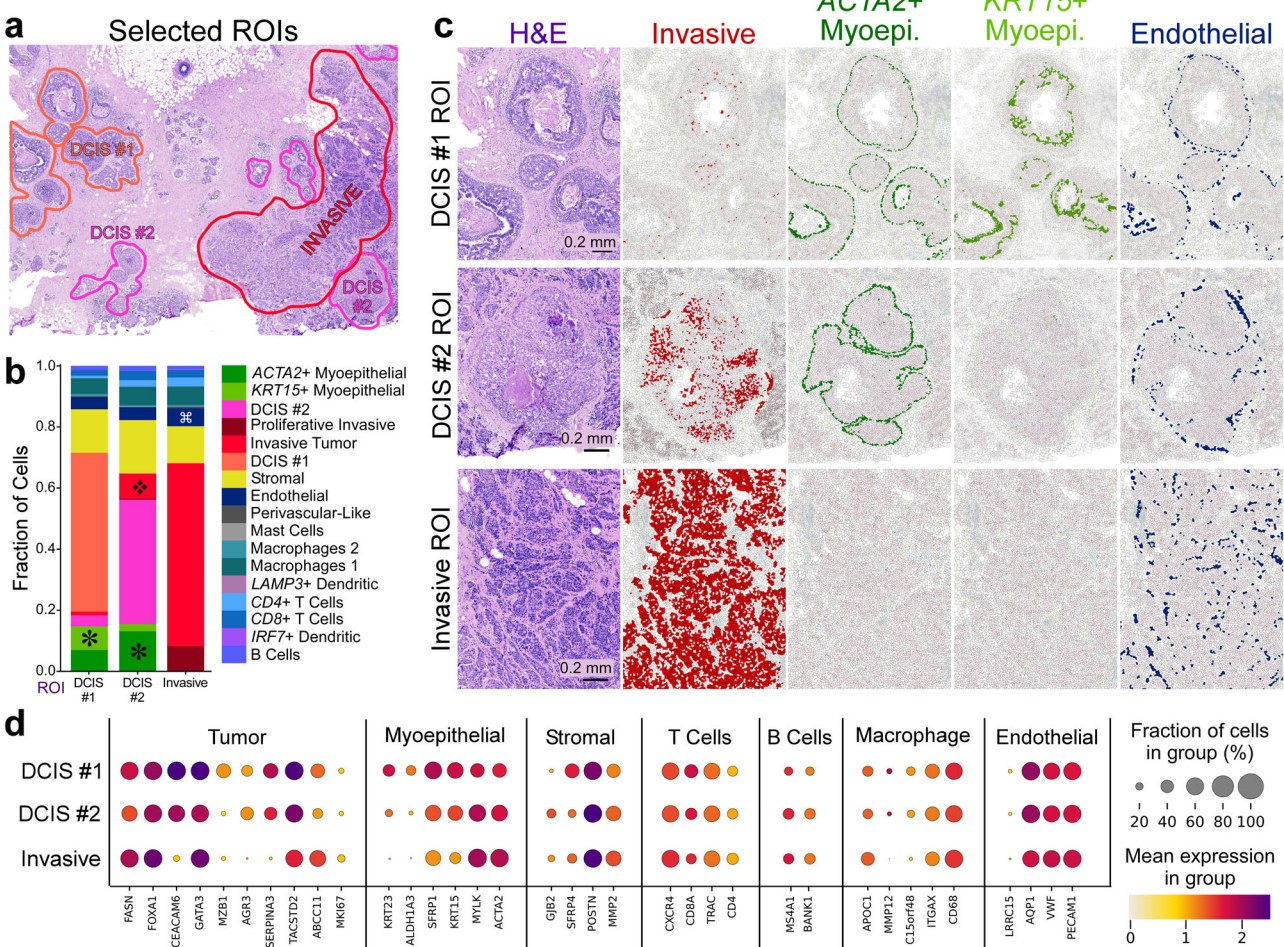

**Fig. 4 | Integrating scFFPE-seq and Xenium data deciphers differences in cell type composition and molecular markers between DCIS subtypes and invasive tumor regions. a** With histology/pathology and scFFPE-seq guidance, we selected three ROIs capturing DCIS #1, DCIS #2, and invasive tumor cell types, and all other cell types in their proximity. **b** We determined the proportions of 17 cell types within these ROIs. We identified four major differences in cell type composition across the ROIs: asterisk = ACTA2+ and KRT15+ myoepithelial cell populations are

distinct in DCIS #1 and DCIS #2 ROIs, but completely absent from invasive tumor ROI; diamond = invasive tumor cells are found within the DCIS #2 ROI; looped square = endothelial cells are found in slightly larger numbers within the invasive ROI. **c** Validation of the finding in (**b**). Scale bar = 0.2 mm. **d** Dot plots showing canonical markers of cell types as well as differentially expressed genes between the tumor subtypes. This experiment was performed in replicate on two serial sections, with one representative section shown here.

our Xenium data. We selected three regions of interest (ROIs): DCIS #1, DCIS #2, and invasive tumor (Fig. 4a). We verified these regions with expert pathologists who observed that (1) post-Xenium H&E exhibited high-quality morphology comparable to standard H&E, and (2) ROIs were either morphologically distinct or were surrounded by a unique

microenvironment. The pathologist annotation showed that DCIS #1 ROI had a smaller and round intermediate nuclear grade ductal hyperplasia with cells that showed mild to moderate variability in size, shape, and placement, and nuclei with variable coarse chromatin. DCIS #2 ROI had invasive carcinoma lesions scattered throughout the

stromal connective tissue, surrounding a large aggregate of highly proliferative ductal carcinoma in situ with two central comedo necrotic formations.

We used scFFPE-seq data to determine proportions of 15 cell types within ROIs in the Xenium data, including lymphocytes, macrophages, stromal, myoepithelial, and invasive cells. We identified four major differences in cell type composition across ROIs (Fig. 4b). ACTA2+ myoepithelial cells were found to be prominent in DCIS #2 ROI, less common in DCIS #1 ROI, and absent in the invasive ROI, invasive tumor cells were found within DCIS #2 ROI, and endothelial cells were found in slightly larger numbers within the invasive ROI. We verified these findings (Fig. 4c) to illustrate how Xenium and scFFPE-seq data can reveal tumor heterogeneity not apparent in the H&E morphology or the pathology report. The DCIS #2 ROI contained many more invasive cells than the DCIS #1 ROI, and also less *KRT15*+ myoepithelial cells, suggesting that DCIS #2 ROI is more invasive than DCIS #1 ROI. The invasive ROI had a high incidence of invasive cell types, and the myoepithelial cell types were entirely absent. The high resolution of Xenium enabled us to capture information about neighboring cells. This is well illustrated in the DCIS #2 ROI with the thin boundary of *ACTA2*+ myoepithelial cells encircling invasive cells (Fig. 4c).

Finally, we graphed the expression of canonical markers representing seven major cell types and differentially expressed genes between the tumor subtypes (Fig. 4d). These analyses revealed that *MZB1* is an exclusive marker of the DCIS #1 ROI and cell type, *GJB2*+ stromal cells were found in the DCIS #2 ROI, *ALDH1A3*, *KRT15*, and *KRT23* were highly expressed in myoepithelial cells of the DCIS #1 ROI, and the macrophage marker *MMP12* was absent from the invasive ROI.

### Integration of whole transcriptome spatial and targeted in situ data provides robust characterization of a small triple receptor positive region

The hormone receptor status of a tumor is important biologically and has clinical relevance. Clinically, breast cancers are classified based on the expression of the estrogen receptor (ER/*ESR1*), progesterone receptor (PR/*PGR*), and human epidermal growth factor receptor 2 (HER2/*ERBB2*)[10]. These classifications typically define treatment strategies; for example, endocrine therapies are commonly used to treat patients with ER+ breast cancers[17]. The tissue block used in this study was annotated as HER2+/ER+/PR−. The Xenium data shows mostly regions of *ERBB2*+ (HER2+) and double positive *ERBB2*+/*ESR1*+ (HER2+/ER+) gene expression (Fig. 5a). However, we identified a small triple positive (*ERBB2*+/*ESR1*+/*PGR*+ (HER2+/ER+/PR+)) DCIS region located in proximity to adipocytes which consisted of a predominantly DCIS #2 tumor epithelium without a *KRT15*+ myoepithelial cell layer (Fig. 5b–d).

Next, we compared expression of the three clinically-relevant receptor genes between the Xenium and scFFPE-seq data (Fig. 5e, f). While few *PGR*+ cells were found in the scFFPE-seq data, these cells did not coexpress *ESR1* or *ERBB2*. In the Visium data, this region is represented by only 5-6 spots (Fig. 5g) as part of Cluster 12 (Fig. 5h), which may have gone unnoticed. However, Visium proved critical here because we could obtain whole transcriptome information from this triple positive region. Using the registration of Xenium to Visium, we binned transcripts from Xenium into the Visium spots by proximity and called this process "spot interpolation" (see "Methods" and Supp. Fig. 10). This allowed us to visualize the cell type proportions within the triple positive region (Fig. 5i), and then performed whole transcriptome differential gene expression analysis of these five spots compared to all other malignant spots. This enabled us to identify 48 differentially expressed genes ($\log_2$FC > 1.5; *p*-value < 0.05) for the PGR+ spots compared to PGR−DCIS #1 and 44 differentially expressed genes for the PGR+ cells compared to PGR−DCIS #2. Four of those genes are shown in Fig. 5j. Next we conducted a gene ontology analysis on the three DCIS categories: triple positive PGR +, PGR−DCIS #1, and

PGR−DCIS #2 to reveal enriched terms from the BioPlanet and Reactome databases (Supp. Fig. 11). We found that the PGR+ spots yielded ontology terms related to ErbB4 and estrogen receptor signaling as well as transmission across chemical synapses. DCIS #1 predominantly showed terms related to metabolism and DCIS #2 was positive for genes related to interferon signaling pathways.

### Integration of single cell and in situ data to profile cells at the myoepithelial boundary

We next wanted to explore whether our data integration strategy could be applied to the study of the DCIS-to-invasive transition in a different biological sample. We obtained a human breast cancer section from a different block (Sample #2) which was annotated as HER2+ and ER − /PR− and contained normal, DCIS and invasive regions. We ran serial 5 μm sections of this sample through the Xenium workflow using the Human Breast Panel described earlier (Supp. Figure 1b). We performed dimension reduction in Seurat and identified immune, myoepithelial, epithelial, and tumor cell populations (Fig. 6a). Adipocytes, cytotoxic T (*CTLA4* +), plasma, dendritic, endothelial, mast, and NK cells were easily detected in this sample as the Xenium gene panel is designed to target these specific cell types. Two stromal populations segregated into distinct clusters and were spatially distinct, with tumor-associated fibroblasts marked by the expression of POSTN, and normal-associated fibroblasts marked by *SFRP4*.

Subclustering analysis of the tumor, myoepithelial, and macrophage populations identified proliferative (*TOP2A* +), *SCD* +, and *S100A8*+ tumor cells, although these populations were not spatially distinct. We were also able to segregate M1, M2 and *CD83*+ macrophages, which are associated with metastasis in breast cancer[18]. Three types of epithelial (*ESR1* +, *PIGR* +, and *OPRPN* +) and one myoepithelial population (*DST* +) were molecularly and spatially distinct, consistent with scRNA-seq reference data where these epithelial subtypes form distinct clusters[4]. A summary of all cell subtypes identified can be found in Supp. Fig. 12. Intriguingly, there were two regions where the histology appeared to have normal duct morphology, but the molecular data revealed tumor cell markers (Supp. Fig. 13). This suggests that the Xenium data can provide insight as to whether a duct will progress towards a carcinoma prior to morphological changes detectable by a pathologist.

We highlight a small population of cells, a subcluster located in between tumor and myoepithelial populations which coexpress tumor (*ERBB2*, *ABCC11*) and myoepithelial markers (*MYLK*, *DST*) (Fig. 6a, a′). High magnification inspection of a region of DCIS with a deteriorating myoepithelial boundary confirmed that these cells express both markers (Fig. 6b, b′, c, c′). As a control, we looked at a normal duct where myoepithelial and epithelial cells are in close proximity, yet cell type-specific markers are not commingled (Fig. 6d, d′) indicating that our observations are not an artifact of gross segmentation errors. Next, we performed differential gene expression analysis on this population of boundary cells identified in the Xenium data (Fig. 6e (red box)). We then looked for this cell type-specific gene expression profile in the scFFPE-seq data from Sample #1 (featured in Figs. 1–5) by matching the expression profile of the rare boundary cells shown in Fig. 6e (red box). Most of these cells had been previously annotated as myoepithelial cells and constitute about ~1% of the cells in the scFFPE-seq data. We compared these cells to tumor and myoepithelial cells and derived whole transcriptome information. This analysis led to the identification of the genes *CX3CL1*, *CCL28*, *PROM1*, and *KLK5*, which were highly expressed in the boundary cells and not in tumor cells, myoepithelial cells or any other colocalized cell type (Fig. 6f).

## Discussion

Resolving the complexities of the tumor microenvironment is necessary for a comprehensive understanding of cancer biology. This is illustrated in our study using an FFPE block from a patient breast

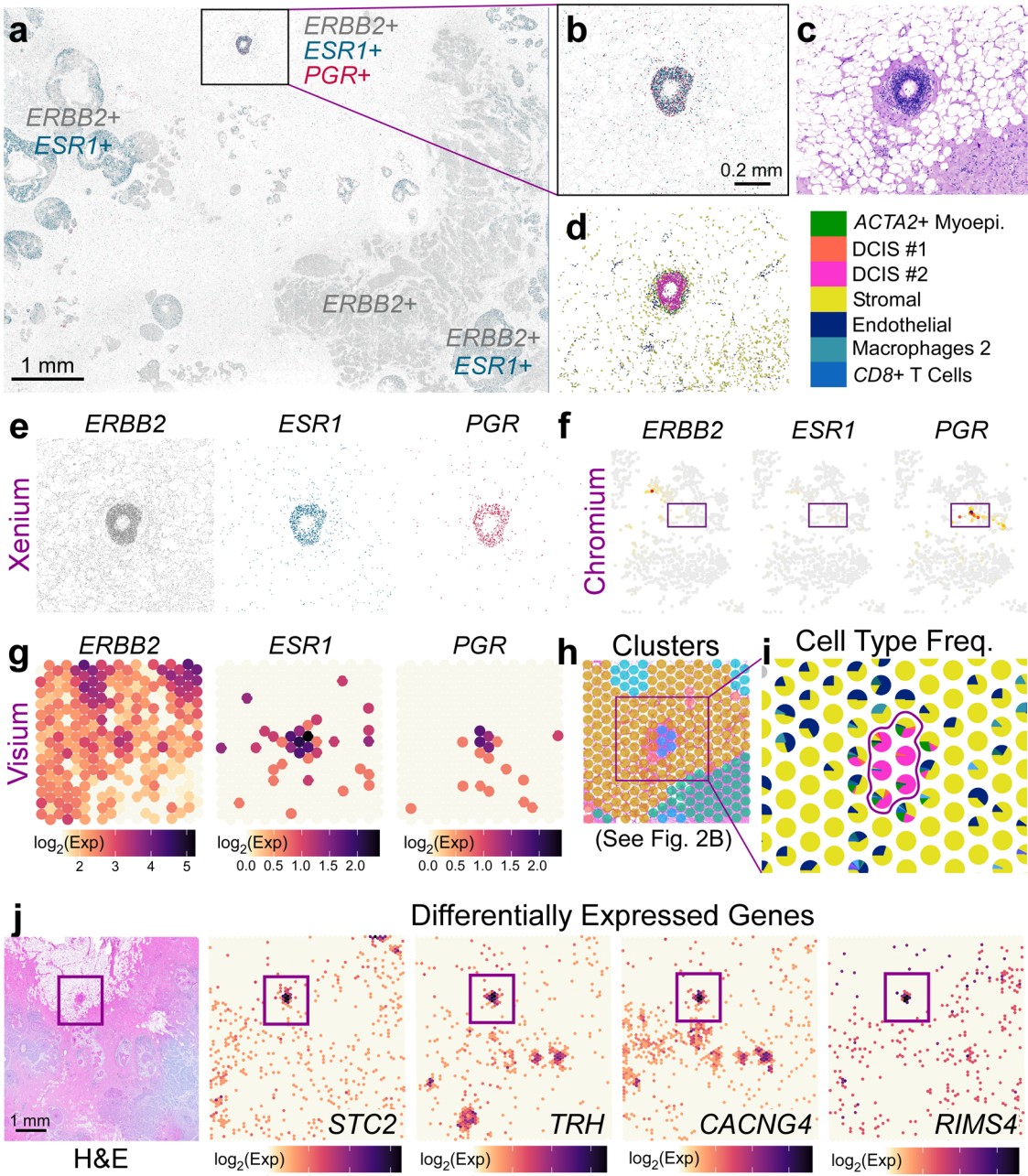

**Fig. 5 | Visium and Xenium integration derive differentially expressed genes in a triple-positive receptor ROI. a** Xenium spatial plot for *ERBB2* (HER2−gray), *ESR1* (estrogen receptor−green), and *PGR* (progesterone receptor−magenta) decoded transcripts. Scale bar = 1 mm. **b** Closer view of triple-positive ROI. Scale bar = 0.2 mm. **c** Corresponding H&E image. **d** Cell types contained within ROI reveal that this is a DCIS #2 tumor epithelium. **e** Individual Xenium spatial plots from (**b**). **f** Chromium scFFPE-seq yields only about 30 cells that are positive for *PGR*, but these cells do not express *ERBB2* or *ESR1*. **g** Triple-positive region is identified in Visium (given a priori knowledge from Xenium) and is **h** part of a distinct cluster (see Fig. 2b). **i** Spot interpolation (see Supp. Fig. 10) provides cell type frequencies within each Visium spot. Color code legend is shown in (**d**). **j** Visium H&E and four representative differentially expressed genes in the tumor epithelium (94 genes; log$_2$FC > 1.5; *p*-value < 0.05) revealed by Visium data across the whole transcriptome. Scale bar = 1 mm. Differential expression was performed in Loupe Browser (see "Methods"), which performs a variant of the negative binomial exact test (for small gene counts), or a fast asymptotic beta test derived from edgeR (for large gene counts). *P*-values were adjusted for multiple testing using the Benjamini–Hochberg procedure to control for the false discovery rate. Both the Xenium and Visium experiments were performed in replicate on two serial sections, with one representative section from each technology shown here.

biopsy that contains both ductal carcinoma in situ and invasive ductal carcinoma. In this model, DCIS refers to neoplastic epithelial cells that remain confined within the ducts, and DCIS is therefore considered nonlethal[15,16,19]. DCIS can be (albeit is not always) the immediate precursor of potentially lethal invasive ductal carcinoma, when the ductal morphology is broken down and cancerous cells invade the stroma. Understanding why some DCIS regions become invasive, while others

do not, remains an open question in the field. Attempts to answer this question often begin with clinical classifications of FFPE blocks by receptor type (e.g., HER2 + /ER + /PR+) and degree of invasiveness and proliferation, but this taxonomy is insufficient to describe heterogeneity within the sample. Despite that the first FFPE block used in our study (Sample #1; Figs. 1–5) was annotated by a pathologist as HER2 + /ER + /PR−, we found a region of DCIS that was positive for the

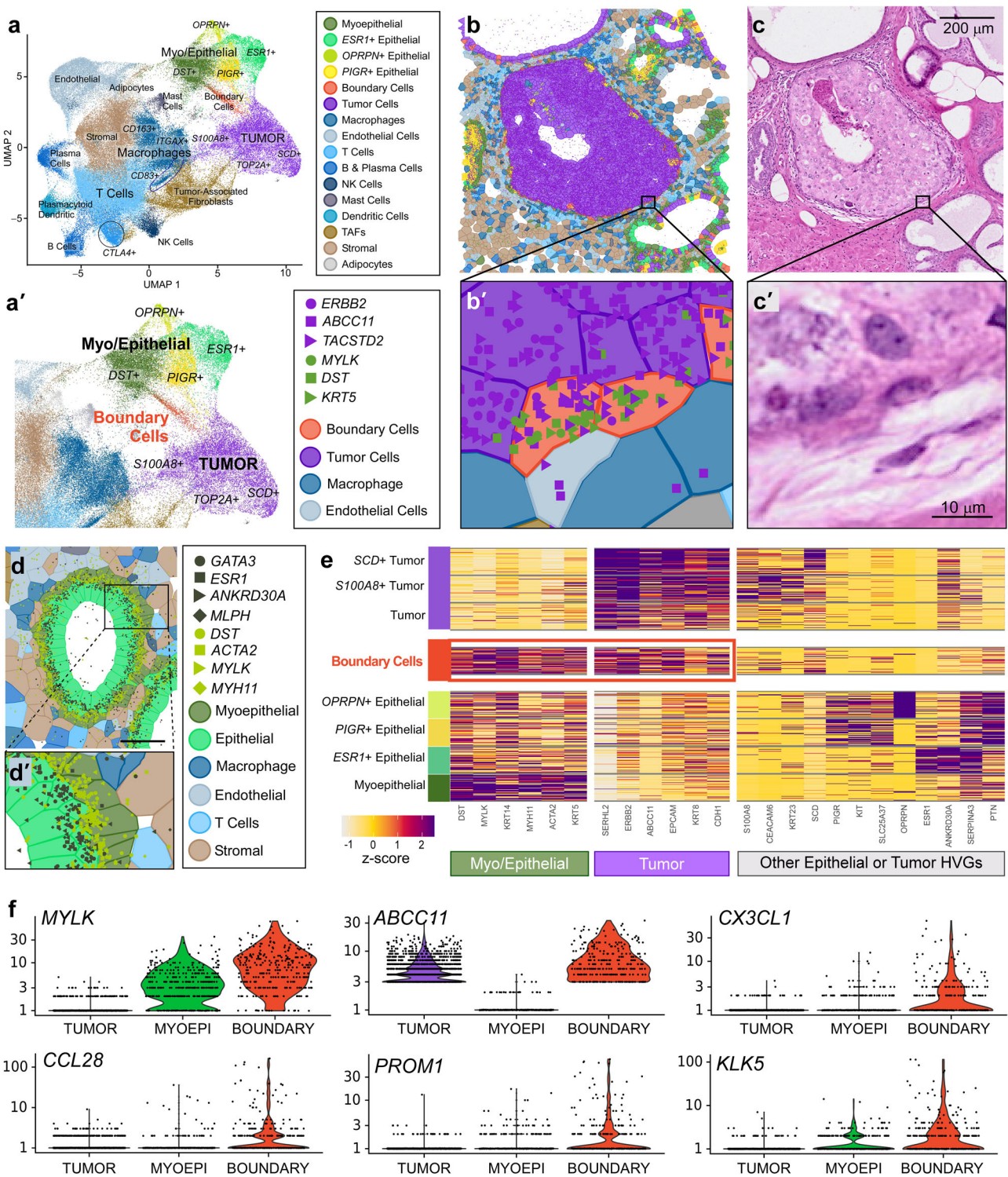

RNA of all three receptors, in just one 5.5 mm × 7.5 mm section of the much larger biopsy (Fig. 5). Furthermore, the block was annotated as 25% DCIS, but this did not capture the fact that at least two molecularly distinct DCIS regions were observed. One DCIS region showed an absence of myoepithelial markers along with the presence of cells already expressing an invasive molecular signature (Fig. 4).

Previous attempts to unmask such tumor heterogeneity, using bulk and single-cell next-generation sequencing (NGS) approaches and immunofluorescence, often target specific genes associated with an invasive/metastatic prognosis and treatment regime[20–22]. Our results are consistent with several published examples, albeit with higher

resolution and gene plexy. For example, low keratin15 (KRT15) expression has been previously suggested to be associated with poor prognosis for patients with invasive carcinoma[21]. Our comparison of two DCIS regions in Sample #1 reveals reduced myoepithelial markers KRT15, KRT23, and ALDH1A3 and increased invasive markers in DCIS #2 (Fig. 4) which could indicate a higher grade tumor. ALDH1A3 (a.k.a. RALDH3), which catalyzes the formation of retinoic acid (RA), is highly expressed and spatially localized to the myoepithelial layer in DCIS #1, but is reduced in DCIS#2 and invasive regions. Although RA has varied roles in cancer, there is some evidence in cell lines that it increases senescence and adhesion to the basement membrane in breast

**Fig. 6 | Chromium and Xenium integration derive differentially expressed genes in a rare cell type. a** Xenium UMAP for a different biological section (different donor) of invasive ductal carcinoma (Sample #2). Most cell types are driven by a single marker (Supp. Fig. 12). TAFs = Tumor-Associated Fibroblasts. When subclustering the epithelial and myoepithelial populations, we noticed a group of cells situated between tumor and DST+ cells, which we label "boundary" and color red. **a′** Zoomed-in view of UMAP from (**a**) showing myoepithelial and epithelial cell subtypes. **b** DCIS ROI containing these cells which are viewed close-up in (**b′**), along with markers for both tumor (purple) and myoepithelial (green) cells. **c** and **c′** Corresponding H&E images. Scale bar = 200 μm in c and 10 μm in (**c′**). **d** Normal duct ROI containing myoepithelial and epithelial cells in closer proximity. **d′** Zoomed in region of (**d**) showing minimal comingling of transcripts representing each cell type: myoepithelial (dark green) and epithelial (light green). Scale bar = 50 μm. **e** Heatmap representation of the UMAP showing relative expression for selected features. HVGs = highly variable genes. Scale bar is a z-score computed across cell types for each gene. Red box highlights that these rare boundary cells express both tumor and myoepithelial markers. The Xenium experiment was performed in replicate on two serial sections, with one representative section shown here. **f** Using the gene expression profile of the rare boundary cells shown in (**e**), we identified this cell type (~283 cells) in the scFFPE-seq data of Sample #1 shown in Figs. 1–5. We conducted a differential gene expression analysis of these cells compared to tumor and myoepithelial cells and validated that these cells express both myoepithelial (*MYLK*) and tumor (*ABCC11*) markers. We further derived genes *CX3CL1, CCL28, PROM1,* and *KLK5* which are differentially expressed in the boundary cells. Differential expression was performed in Loupe Browser (see "Methods") which performs a variant of the negative binomial exact test (for small gene counts), or a fast asymptotic beta test derived from edgeR (for large gene counts). *P*-values are adjusted for multiple testing using the Benjamini–Hochberg procedure to control for the false discovery rate.

myoepithelial cells, thereby decreasing the invasive capacity of tumor epithelium[23]. Xenium identified *AGR3* as a tumor epithelial marker associated with DCIS ROIs, but not the invasive ROI (Fig. 4). *AGR3* protein expression in breast tumors is significantly associated with estrogen receptor α and lower tumor grade[22], suggesting that *AGR3* could serve as a biomarker for prognosis and early detection.

In a second biological sample of invasive ductal carcinoma (Sample #2), we identified regions of the tissue that had normal ductal morphology, as annotated by a pathologist based on the H&E staining. However, the molecular data revealed several ducts containing tumor cells. It has been reported that pathology diagnoses are only concordant 75% of the time[24]. The data we present here demonstrates what early diagnosis would potentially look like, detecting the cancer before the morphology is drastically altered. Thus, this ability of Xenium to map the localization and expression level of key genes at high resolution holds great promise to transform, diagnose, prognose and guide more effective treatment management.

Female breast cancer is the most commonly diagnosed cancer globally, with ~2.3 million new cases reported in 2020[25], and the prevalence of cancer as a leading cause of premature death is ever-increasing worldwide, particularly in developing nations[26]. How can single-cell, spatial, and in situ technologies scale to deal with this challenge? The key to this question is the ability of these technologies to glean high quality data from FFPE tissues. FFPE methods for preservation of samples are well established in clinical practice as they allow for a high degree of morphological detail to be maintained, and as such, there are large numbers of FFPE specimens in biobanks that are potentially available for genomics research[27]. Using RTL technology with carefully designed probe sets, we are able to overcome the formalin-induced obstacles of strand cleavage and cross-linking that have plagued researchers for decades.

In the present study, we used three independent but complementary genomic technologies to explore the biology of FFPE-preserved tissue sections. Our results typify how the integration of these technologies is an iterative process, and suggest how discoveries in one data modality can rapidly inspire explorations in another. What did each technology bring to the table, and what did we learn from integrating them that we could not have learned from a single technology individually? scFFPE-seq is the most sensitive of the three, particularly for lowly expressed genes. We found that scFFPE-seq median gene sensitivity was higher than both Chromium 5′ and 3′ GEX data, from patient-matched dissociated tumor cells (Supp. Fig. 7). Of the three FFPE-compatible assays presented here, scFFPE-seq is the only one offering whole transcriptome data at single-cell resolution, making it well suited for establishing a baseline of disease, annotating cell types (Fig. 2a), and designing or validating targeted Xenium gene panels. Like scFFPE-seq, Visium also provides whole transcriptome data. Although Visium lacks true single cell resolution at this time, it provides a spatial context that cannot be explored with single-cell

technologies. Integrating scFFPE-seq and Visium data was straightforward due to the identical probe set used in both technologies, and allowed for accurate deconvolution of cell types that composed the Visium spots (Supp. Fig. 10).

In the early stages of data exploration, Visium and H&E data were used to annotate three tumor cell types within the scFFPE-seq data by noting that the differentially expressed genes in specific scFFPE-seq clusters were mapping to the invasive tumor domain, or one of two spatially distinct DCIS regions in the H&E image (Fig. 2). Hence, Visium alone identified that there were three spatially distinct tumor subtypes, which was not captured in the pathologist annotations. When we clustered Xenium data from Sample #1 without relying on the Visium and scFFPE-seq derived cell annotations, we did not resolve the two distinct DCIS cell types, further substantiating that information is gained by using an integrative approach. We then integrated Xenium and Visium data to derive differentially expressed genes from a tumor region containing cells expressing RNA of three receptors (ERBB2 +/ESR1 +/PGR+). Neither Visium nor scFFPE-seq identified these cells initially because they were so sparse, and their detection required the high-resolution spatial information gained by Xenium. Using spot interpolation methods (Supp. Fig. 10), we identified the relevant tumor epithelial cells in the Visium data and derived whole transcriptome information. Gene ontology analysis revealed that the triple positive (*ERBB2 +/ESR1 +/PGR+*) spots express the receptor *ERBB4* (a.k.a. HER4) and its cognate ligand *EREG* (a.k.a. epiregulin) in addition to *ESR* and *PGR*. This is a reasonable finding given that HER4 is correlated with ER and PR expression in breast cancer, and when all of these receptors are found together, the prognosis can be more favorable (reviewed in ref. 28). Although the triple positive region is small in this particular section, it might be representative of a larger percentage of cells elsewhere within the biopsy and potentially change the classification and treatment protocol.

Xenium is particularly suited for investigations of intricate tissues with a high diversity of cell types in a small area, including immune and myoepithelial cells (Figs. 4–6), that elude lower resolution technologies. One example of this was the identification of a small population of boundary cells in Sample #2 (Fig. 6), which were discovered by subclustering the epithelial and myoepithelial cell clusters. We were then able to identify cells co-expressing both tumor and myoepithelial markers in the scFFPE-seq data (from Sample #1). Differential gene expression analysis identified *CX3CL1, CCL28, PROM1,* and *KLK5* as being upregulated in this cell population. *CX3CL1* promotes breast cancer metastasis[29,30] and *CCL28* is an epithelial, tumorigenic cytokine[31]. *PROM1* (a.k.a. *CD133*) is a well-characterized cancer stem cell marker[32]. It is an open question in the field whether myoepithelial cells during the DCIS-to-invasive transition commit apoptosis, dedifferentiate, senesce, or transdifferentiate. Our findings are in good agreement with a recent study that highlighted that some tumor-associated myoepithelial cells express oncogenic cytokines and

promote invasiveness, contrary to normal myoepithelial cells which are considered protective[33]. This suggests that the existence of a transitional boundary cell which exhibits properties of both myoepithelial and tumor cells is very possible, however, further investigation would be required in order to validate this.

High-resolution in situ analysis of complex tissues will revolutionize how we understand biology, providing insights not previously possible with other technologies. Integration of the data we present here with imaging mass cytometry and CITE-seq data, to further extract biological insights has already been performed[34]. With the continued development of tools to study these biological questions, we will derive an even greater understanding of molecular profiles as they relate to the tissue architecture, and how cells interact with other cells and non-cellular components in their local tissue environment. Our findings demonstrate that the highest resolution and richest biological information are gleaned through the combination of complementary technologies. While each technology independently elucidates high-quality gene expression data from FFPE tissues, it is the integration of this data that illuminates biology with more rigor and refinement than with a single technology alone. The resolution and breadth of the technologies we describe have promising implications across the biological sciences, but particularly in the future of translational and clinical research, and ultimately, in advancing human health.

## Methods

### Samples and sample collection

*Sample #1*. A single formalin-fixed, paraffin-embedded (FFPE) breast cancer tissue block (TNM stage T2N1M0, ER + /HER2 + /PR −) was collected on 2021-07-26 and obtained from Discovery Life Sciences. Corresponding dissociated tumor cells, fresh frozen in liquid nitrogen, were also sampled from the same biopsy (patient matched). 5 µm sections were taken from the FFPE tissue using a microtome (Thermo Scientific HM355S; MX35 blades). For the Chromium Single Cell Gene Expression Flex (scFFPE-seq) workflow, 25 µm FFPE curls were collected into a tube prior to serial sectioning for Visium CytAssist and Xenium (two replicates of 5 µm sections for each spatial platform), then an additional 25 µm FFPE curl was collected into the same tube reserved for scFFPE-seq. These pooled 25 µm curls (50 µm total) were treated as a single replicate. Another replicate could not be performed due to the large amount of input material required by scFFPE-seq and needing to reserve the same block for multiple technologies.

*Sample #2*. A formalin-fixed, paraffin-embedded (FFPE) breast cancer tissue block (AJCC pathologic stage pT2 pN1a pMX, ER − /HER2 + /PR −) was collected on 2009-07-24 and obtained from Discovery Life Sciences. 5 µm sections were taken from the FFPE tissue using a microtome (Thermo Scientific HM355S; MX35 blades).

### Chromium 3′ and 5′ single-cell gene expression (GEX)

We collected Chromium 3′ and 5′ GEX data from dissociated tumor cells to benchmark performance against the scFFPE-seq data. Dissociated tumor cells were recovered following Demonstrated Protocol CG000233. For the 3′ and 5′ workflows, cells were loaded on to the Chromium X instrument following the library preparation protocols in the Chromium Next GEM Single Cell 3′ Reagent Kits v3.1 User Guide (CG000204) and Chromium Next GEM Single Cell 5′ Reagent Kits v2 (Dual Index) User Guide (CG000331), respectively. Libraries were sequenced on an Illumina NovaSeq with paired-end dual-indexing (28 cycles Read 1, 10 cycles i7, 10 cycles i5, 90 cycles Read 2). All of the 3′ and 5′ flowcells were demultiplexed with bcl2fastq (Illumina). FASTQ files were processed with Cell Ranger v7.0.1 (10x Genomics), using the cellranger count pipeline on each GEM well with the GRCh38-2020-A reference to produce gene-barcode matrices and other output files, followed by aggregation of GEM wells with the cellranger aggr pipeline.

### Chromium Single Cell Gene Expression Flex (scFFPE-seq)

Our goal in producing scFFPE-seq data was to precisely define the cell types present in serial tissue sections to enable downstream integration of data types. 50 µm FFPE curls were dissociated with the Miltenyi Biotech FFPE Tissue Dissociation Kit. Approximately 600,000 cells were washed, counted, and resuspended, loading 16,000 cells per each of four GEM wells (targeting 10,000 recovered cells) on a single Chromium X chip. Sequencing libraries were generated following the User Guide (CG000477). Libraries were sequenced on an Illumina NovaSeq with paired-end dual-indexing (28 cycles Read 1, 10 cycles i7, 10 cycles i5, 90 cycles Read 2). Sequencing libraries were demultiplexed with bcl2fastq (Illumina). FASTQ files were processed with Cell Ranger v7.0.1 (10x Genomics) using the multi pipeline and the GRCh38-2020-A reference.

### Visium CytAssist

**Whole transcriptome spatial data**. Our goal in producing Visium CytAssist data was to obtain whole transcriptome, spatially-barcoded sequence data in serial sections. The histology workflow was performed using the Visium CytAssist Spatial Gene Expression for FFPE (Demonstrated Protocol CG000520). The tissue was sectioned as described in Visium CytAssist Spatial Gene Expression for FFPE – Tissue Preparation Guide (Demonstrated Protocol CG000518). 5 µm sections were placed on a Superfrost™ Plus Microscope Slide (Fisherbrand™) and H&E-stained following deparaffinization. Sections were imaged, decoverslipped, followed by hematoxylin destaining and decrosslinking (Demonstrated Protocol CG000520). The glass slide with tissue section was processed with a Visium CytAssist instrument to transfer analytes to a Visium CytAssist Spatial Gene Expression slide with a 0.42 cm² capture area. The probe extension and library construction steps follow the standard Visium for FFPE workflow outside of the instrument. Libraries were sequenced with paired-end dual-indexing (28 cycles Read 1, 10 cycles i7, 10 cycles i5, 90 cycles Read 2). Sequencing libraries were demultiplexed with bcl2fastq (Illumina). The Space Ranger pipeline v2022.0705.1 (10x Genomics) and the GRCh38-2020-A reference were used to process FASTQ files.

### Xenium in situ workflow

**Gene panel design**. The Xenium In Situ technology uses targeted panels to detect gene expression. 313 genes for cell type identification (280 of which are included in the Xenium Human Breast Panel) were selected and curated primarily based on single-cell atlas data for human breast tissue[4,13,14]. The probes were designed to contain two complementary sequences that hybridize to the target RNA and a third region encoding a gene-specific barcode, so that the paired ends of the probe bind to the target RNA and ligate to generate a circular DNA probe. If the probe experiences an off-target binding event, ligation should not occur, suppressing off-target signals and ensuring high specificity.

**Xenium sample preparation**. The Xenium workflow (using in-development chemistry and a prototype instrument and consumables) began by sectioning 5 µm FFPE tissue sections onto a Xenium slide, followed by deparaffinization and permeabilization to make the mRNA accessible. The mRNAs were targeted by the 313 probes described above and two negative controls: (1) probe controls to assess non-specific binding and (2) genomic DNA (gDNA) controls to ensure the signal is from RNA. Probe hybridization occurred at 50 °C overnight with a probe concentration of 10 nM. After stringency washing to remove un-hybridized probes, probes were ligated at 37 °C for two hours. During this step, a rolling circle amplification (RCA) primer was also annealed. The circularized probes were then enzymatically amplified (for one hour at 4 °C followed by two hours at 37 °C), generating multiple copies of the gene-specific barcode for each RNA binding event, resulting in a strong signal-to-noise ratio.

After washing, background fluorescence was quenched chemically. The biochemistry is designed to mitigate autofluorescence, which is a known issue due to the presence of lipofuscins, elastin, collagen, red blood cells, and formalin-fixation itself[35]. Sections were placed into an imaging cassette to be loaded onto the Xenium Analyzer instrument.

**Xenium analyzer instrument.** The Xenium Analyzer is fully automated and includes an imager (imageable area of about 12 × 24 mm per slide), sample handling, liquid handling, wide-field epifluorescence imaging, capacity for two slides per run, and an on-instrument analysis pipeline. The imager is a fast area scan camera featuring a high numerical aperture, a low read noise sensor, and ~200 nm per-pixel resolution. On the Xenium Analyzer, image acquisition was performed in cycles. The reagents, including fluorescently labeled probes for detecting RNA, were automatically cycled in, incubated, imaged, and removed by the instrument. Following the binding of fluorescent oligos to the amplified barcode sequence, the sample underwent 15 rounds of fluorescent probe hybridization, imaging, and probe removal. The Z-stacks were taken with a 0.75 μm step size across the entire tissue thickness.

**Image pre-processing.** The Xenium Analyzer captured a Z-stack of images every cycle and in every channel, which needed to be processed and stitched to build a spatial map of the transcripts across the tissue section. Stitching was performed on the DAPI image, taking all of the stacks from different FOVs and colors to create a complete 3D morphology image (morphology.ome.tif) for each of the stained regions. First, the lens distortion in internal sensor data was corrected based on instrument calibration data, which were collected in order to characterize the optical system and were saved on-instrument. Next, the Z-stacks from the internal sensor data were further subsampled to a 3 μm step size, which was determined empirically to be a useful resolution for cell segmentation quality. Image features were then extracted from the regions where FOVs overlapped. Feature matching was performed to estimate the offsets between adjoining FOVs. The offsets were used to ensure consistent global alignment across the image. Finally, the 3D DAPI image volumes (Z-stacks) generated across FOVs were stitched together.

**RCA product image processing.** The goal of RCA product image processing was to detect and filter puncta and correct distortion. A punctum is a point source in microscopy, smaller than a pixel, and is measured in units of observed photons. The 3D image volumes (Z-stacks) obtained for each FOV were processed, for four color channels and 15 cycles, to detect the puncta in 3D space that correspond to labeled RCA products. The RNA fluorescence images were scanned for punctum signals that stand out from the local background. The XYZ coordinates of each punctum were refined by examining local brightness. The signal intensity of the punctum was determined by fitting a Gaussian distribution to the observed emitted light to determine the center, size, and intensity of the point sources. We filtered out puncta that were unlikely to be from true RCA products (non-punctate or low-quality signals). Similar to DAPI images, curvature distortion was corrected.

**Decoding.** In order to proceed from puncta to transcripts, decoding was performed using a Xenium codebook—a collection of codewords that were assigned to genes in the gene panel (gene_panel.json). Each codeword was defined based on an expected pattern of fluorescent signals recorded across channels and cycles. Some codewords were reserved for negative controls. The fluorescent signals from all channels and cycles were compared to the codebook using a global (across all FOVs) maximum likelihood approach. This approach considered attributes such as puncta locations, their color and cycle of detections, and signal intensities.

**Q-Scores and controls.** A Phred-style calibrated quality score (Q-Score) was assigned to each decoded transcript to signify the confidence in the decoded transcript identity. Raw Q-Scores were derived from the likelihood of the maximum likelihood codeword (the codeword that best explains the observed data) compared to the likelihood of other sub-optimal codewords. Codewords were mapped to targets using the gene panel information. Final Q-Scores were calculated by first binning the full range of raw Q-Scores, then the raw Q-Scores in each bin were calibrated by the proportion of "Negative Control Codewords" in the bin. A final Q-Score was assigned to each bin to ensure that each bin's Q-Scores were correctly calibrated. Control probes were built into the process to ensure that the final Q-Scores were accurately calibrated. Three types of controls were used:

1. *Negative control codewords* are codewords in the codebook that do not have any probes matching that code. They are chosen to meet the same requirements as regular codewords and can be used to assess the specificity of the decoding algorithm.
2. *Negative control probes* are probes that exist in the panels but target non-biological sequences. They can be used to assess the specificity of the assay.
3. *Unassigned codewords* are unused codewords. There is no probe in this particular gene panel that will generate the codeword.

We only included transcripts with a Q-Score ≥ 20 in the cell-feature matrix and downstream analyses.

**Cell segmentation.** In order to assign mRNA transcripts to cells, we first segment nuclei based on the signal in the DAPI morphology image and then assign transcripts to the closest nucleus within a maximum distance of 15 μm. Transcript assignment was performed using a 2D segmentation mask that was the result of combining multiple 2D segmentations taken at different Z-planes. Individual nuclei were detected and nuclei that were close in X, Y, and Z were identified as a single nucleus. The final 2D segmentation did not allow for overlapping nuclei.

DAPI-based nucleus segmentation was achieved using a deep-neural-network approach. This approach is conceptually similar to the popular CellPose algorithm[36] in that an encoder-decoder neural network does not directly solve for the segmentation mask, but instead solves a related problem that is easier to learn and allows for the imposition of geometric constraints. The training data is based on hundreds of thousands of hand-drawn cells covering a wide range of tissues imaged on the Xenium instrument. Segmentation quality was judged by holding back a subset of hand-labeled images for benchmarking purposes. We adopted the evaluation methodology used by Greenwald et al., 2021[37] and first proposed in Moen et al., 2019[38]. We trained to achieve an F1 score of greater than 0.80 on benchmark datasets using an overlap threshold of 0.5 for detection.

**Output file export.** A variety of output files were produced by the on-instrument pipeline. The essential files used downstream were the feature-cell matrix (HDF5 and MEX formats identical to those output by Cell Ranger and Space Ranger for Chromium and Visium data, respectively), the transcripts (listing each mRNA, its 3D coordinates, and a quality score), and the cell boundaries CSV file. These files were then transferred for downstream analysis off-instrument.

### Post-Xenium histology
**H&E and IF staining.** The post-Xenium H&E staining followed Demonstrated Protocol CG000160. For post-Xenium IF staining, sections were washed with PBST, then incubated in a blocking buffer (ScyTek AAA999) for 30 min at room temperature. The primary antibody (Table 1) in the blocking buffer was added and incubated in the dark at 4 °C overnight. The following day, the sections were washed three times (10 min each) with PBST then incubated with secondary

**Table 1 | Antibodies used for the post-Xenium IF staining**

| Protein | Host | Fluor | Vendor | Catalog | Dilution |
|---------|------|-------|--------|---------|----------|
| CD20 | Mouse | Unconj. | Abcam | AB219329 | 1:200 |
| HER2 | Rabbit | Unconj. | Abcam | AB134182 | 1:1500 |
| Ms IgG | Goat | Alexa 488 | ThermoFisher | A-11029 | 1:500 |
| Rb IgG | Goat | Alexa 594 | Abcam | AB150088 | 1:500 |

antibodies (Table 1) and DAPI in a blocking buffer, in the dark at room temperature, for two hours. Next, the sections were washed three times (10 min each) with PBST. Sections were imaged in a proprietary Xenium imaging buffer and imaged on a Zeiss Axioimager with a 40x dipping objective. The Zen Blue software was used for tiling, image acquisition, and exporting TIFF files. Post-Xenium H&E and IF images were registered to Xenium data using Fiji BigWarp.

**Downstream analysis & integration**
**Chromium & Visium post-processing.** The 3′, 5′, and scFFPE-seq data were filtered with scanpy 1.19. Cell filtering parameters included largest gene fraction ≤0.2, mitochondrial fraction ≤0.15, and number of genes observed ≥500. We performed t-distributed stochastic neighbor embedding (t-SNE) on Chromium and Xenium data using the monet package v0.3.2[39]. A principal component analysis (PCA) was performed on the feature-cell matrix, and the top 50 components were input to the t-SNE. We subsampled the whole transcriptome Chromium and Visium data to only the 313 genes used in the Xenium panel. The Xenium t-SNE coordinates were initialized with the scFFPE-seq cluster centers. The Visium t-SNE was generated in Loupe, and expression is either reported as log2(counts) when shown stand-alone, or as raw counts when comparing directly with Xenium transcript counts.

**Supervised labeling & label transfer.** We annotated the scFFPE-seq data by first conducting a differential gene expression (DGE) analysis across unsupervised clusters in Loupe. Annotations were built upon this DGE analysis, and literature review[4,10], and pre-CytAssist H&E staining. We assigned labels DCIS #1 and DCIS #2 according to transcriptional similarity between the scFFPE-seq and Visium platforms. We performed a log-normalization step of the data, and then calculated a z-score across cells. A PCA was performed and the top 50 PCs were selected. From the in situ data, we determined the 30 nearest neighbors for each cell after normalization and projection into PC space, and if at least 50% were one cell type, then that is the cell type that was assigned. If that criteria was not met, then the cell was classified as "unlabeled".

**Unsupervised labeling and subclustering.** For Sample #2 (Fig. 6 and Supp. Fig. 13), we used the Seurat vignette (https://satijalab.org/seurat/articles/spatial_vignette_2.html) as guide to load and analyze the Xenium data with the development branch of Seurat 5 (https://github.com/satijalab/seurat/tree/develop). We identified 14 clusters (resolution = 0.3), and further subclustered the epithelial and macrophage clusters to increase resolution of the cell types within. Annotations were aided by single cell atlas data[4,13,14]. Cropped FOV images with segmentation were generated with the ImageFeaturePlot function, and a custom ggplot2 script was used to plot individual transcripts on top of the segmented cells.

**Chromium and Visium differential gene expression (DGE).** Cell Ranger, Space Ranger, and Loupe Browser test, for each gene and each cluster, whether the in-cluster mean differs from the out-of-cluster mean using one of two methods. When gene counts are small, the quick and simple method, a version of the negative binomial exact test[40] was used. For larger counts, a modified version of the fast asymptotic beta test[41] was used. For the differential gene expression

analysis shown in Fig. 6, we created a heatmap in Seurat v4.3 and obtained variable features using the vst selection method. These features were used to define a gene expression profile for tumor, myoepithelial, and rare boundary cells. We then used the Filter function in Loupe Browser v6.4.1 to threshold marker genes and assign identities (tumor, myoepithelial, and boundary) to barcodes in the scFFPE-seq data. We then performed a locally distinguishing feature comparison to find novel differentially expressed genes in the rare boundary cells. We visualized these genes using VlnPlot in Seurat v4.3.

**Gene ontology analysis.** Using spot interpolation (see Supp. Fig. 10), we derived information about the cell composition of the triple positive region (ERBB2 + /ESR1 + /PGR +). Using Loupe Browser v6.4.1, we lassoed around the spots within the triple positive domain that were predominantly composed of the DCIS cell type. We then conducted a global differential gene expression analysis to derive genes featured in Fig. 5j. For the gene ontology analysis, we used Loupe to lasso around PGR−DCIS #1 and PGR−DCIS #2 cells, and compared them to the triple positive region (PGR +). We then took the differentially expressed genes and inputted them into Enrichr to obtain the ontology information shown in Supplemental Fig. 11.

**Xenium differential gene expression (DGE).** We drew a region of interest (ROI), a polygon around morphological features (individual cells, groups of cells, etc.) and performed DGE across these ROIs with scanpy v1.19. ROI selection was performed in the Xenium Explorer software (development version, 10x Genomics), and significance was assessed with the Wilcoxon test on log-normalized count data. The DGE was performed for each cell type across ROIs.

**Benchmarking sensitivity.** The Xenium assay's sensitivity is unique for each gene. Therefore, we designed probes to estimate the effective sensitivity of the assay for each gene, and we describe the effective sensitivity as an average or median gene sensitivity. Because mean sensitivity is biased by high expressors, we calculated median gene sensitivity by first computing the sensitivity of each gene separately (the mean of the counts per cell), then calculating the median across all genes. Because sensitivity is dependent on sequencing saturation, the 3′ and 5′ GEX data were downsampled to 10,000 mean reads per cell to match the sequencing depth of 10,000 reads per cell (the recommended depth) for scFFPE-seq, and 20,000 mean reads per cell (the recommended sequencing depth for the 3′ and 5′ assays). The 3′ and 5′ GEX data were also downsampled to only the genes on the RTL scFFPE-seq probe set.

**Image registration.** For registration of IF images to the Xenium morphology images, which are both DAPI images, we used a SIFT registration with the cv2 4.5.4 package in python v3.9.7, which produces the transformation between IF and Xenium (see the 10x Analysis Guide available at https://www.10xgenomics.com/resources/analysis-guides/he-to-xenium-dapi-image-registration-with-fiji). For registration of Visium to Xenium data, serial sections were rotated 2.58 degrees relative to each other, then a manual-defined keypoint registration between the corresponding H&E images (serial sections) was used. Over 100 landmark features were identified on commonly shared microstructures. Using RANSAC, we determined the subset of coordinates that matched, and performed the transformation between coordinates with the FindHomography() function in the cv2 package.

**Visium/Xenium spot interpolation and deconvolution.** Using the registration of Xenium to Visium, we binned cells (by centroid) and transcripts from Xenium into the Visium spots. This was done by proximity. The closest spot to a cell or transcript was identified as the spot a cell or transcript lies within. Robust Cell Type Decomposition (RCTD) with spacexr 2.0.1[42] in R was used to deconvolve Visium spots into cell types using the unsupervised scFFPE-seq reference. See the 10x

Analysis Guide (https://www.10xgenomics.com/resources/analysis-guides/integrating-10x-visium-and-chromium-data-with-r).

## Reporting summary

Further information on research design is available in the Nature Portfolio Reporting Summary linked to this article.

## Data availability

The raw scFFPE-seq, Visium, and Xenium data generated in this study have been deposited in the GEO database under accession code GSE243280. The data are available here: *Downloadable datasets:* https://www.10xgenomics.com/products/xenium-in-situ/preview-dataset-human-breast. *Interactive data explorer:* https://www.10xgenomics.com/products/xenium-in-situ/human-breast-dataset-explorer. In addition, source files have been uploaded which provide raw data for the heatmaps, violin plots, and bar graphs in the Article. Source data are provided with this paper.

## Code availability

The following software programs used for analysis and visualization have been previously reported and are made available by 10x Genomics, Inc. ("10x") to the public, and current versions can be downloaded from https://www.10xgenomics.com/support#software: Cell Ranger, Space Ranger, Xenium Ranger, Loupe Browser, and Xenium Explorer (all versions thereof collectively, "Software"). You may download and use the Software solely with data generated using 10x products, including with the data provided by the authors in connection with this manuscript, as detailed in the 10x End User Software License, available at https://www.10xgenomics.com/legal/end-user-software-license-agreement. To summarize, the right to use Software is limited, non-exclusive and revocable, and no right is granted to sublicense, transfer, or distribute the Software to a third party, nor to permit access to or use of the Software by any third party. Merging, combining, modifying, or reverse engineering the Software is strictly prohibited. Custom scripts used for this paper that are not Software (e.g., for spot interpolation) ("Code") are available on GitHub at https://github.com/10XGenomics/janesick_nature_comms_2023_companion, under the non-exclusive license provided with the Code. You may use the Code solely with data generated using 10x products, including for the purpose of validating the data and results of this paper, and you may not redistribute, license or sublicense the Code to any third party without 10x's prior written permission. Any breach or attempt to circumvent the use restrictions above will automatically terminate all your rights to use Software and Code under the licenses granted. The Software and Code are provided by 10x "AS IS" without warranties of any kind, statutory, express or implied, including but not limited to the warranties of merchantability, fitness for a particular purpose, or noninfringement. 10x is not responsible for any use by you of the Software or Code, including any damages related to your use of the Software or Code.

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

## Acknowledgements
We would like to thank the Consortium of 10x Development Teams who built the products used in this manuscript and the 10x Leadership Team for their support in this work. We also thank the 10x Genomics microscopy, sequencing, and histo-pathology cores.

## Author contributions
A.J., R.S., and S.T. conceived the study; A.J., G.B., C.M., M.O., A.K., J.A., T.D., and S.M. performed experiments; A.J., R.S., F.W., S.W., M.R., J.S., and S.T. analyzed and interpreted data; A.J., R.S., A.G., and S.T. wrote the manuscript.

## Competing interests
All authors are current or previous employees of 10x Genomics.

## Additional information

## 10x Development Teams

Jawad Abousoud[1], Tingsheng Yu Drennon [1], Andrew Kohlway[1], Robert Shelansky[1], Jordan T. Sicherman [1], Sarah E. B. Taylor [1] ✉, Florian Wagner [1] & Stephen R. Williams[1]

A full list of members and their affiliations appears in the Supplementary Information.

