## [Peer Review File · Nature Communications]

High resolution mapping of the tumor microenvironment using integrated single-cell, spatial and in situ analysisReviewers' Comments:

Reviewer #2:

Remarks to the Author:

The authors present a combination of technologies, including a new, commercially available Xenium technology, the existing commercial technologies Visium and Chromium. They apply these technologies to investigate an FFPE block of human breast cancer. While the technology is clearly impressive and the opportunities that arise with combining these three technologies, the study stays at a very superficial, exploratory level, mainly discussing advances - without comparing them to other current available methods, nor showing a concrete example of how these technologies can be used to advance medical research (as they claim in the discussion)

Overall, the paper reads more like an advertisement of the technologies, than a study that has a clear biological goal. This is of course fine, since technology development is crucial for advancing biology, however in this particular case, all the technologies are commercially available, which leaves the reader wondering what is the novelty of this.

- Can the authors point out how more clearly how the technologies they combined (each of them commercially available) can generate real biological insights?
- the authors should also point out the limitations of the technology and where appropriate mention alternative technologies
- If the goal is to describe the technology, this should be done in more detail, also seriously pointing out limitations, competing technologies
- if the goal is to describe an application of the technology, the biological question should be more convincing - and exploratory analysis of just one breast FFPE block of one breast cancer sample can hardly be the way to learn something about breast cancer in general. There are some interesting leads in the paper that could be further explored, such as the detection of adipocytes
- if the goal is to present a data integration strategy, more focus should be on the data integration algorithms, which are currently not much explained in the main text.

Overall, great technology, yet the fact that it is already commercially available makes it of course less novel, so I would think a more convincing biological question would be more appropriate to demonstrate the power of the combination of the proposed technologies.

Reviewer #3:

Remarks to the Author:

The critical study by the Taylor group presents two novel technologies, in situ hybridization platform (Xenium In Situ) and the advanced single-cell gene expression workflow for FFPE samples (scFFPE-seq). Albeit lacking whole transcriptome profiling, Xenium provides higher gene detection sensitivity with subcellular spatial resolution. Integration of these technologies with the previous 10x Visium would be extremely powerful to explore deeper mechanisms of development and disease. For example, the authors apply them to serial breast cancer FFPE sections and find the molecular and cellular differences of tumor regions with and without invasion potential. These new platforms are exciting and promising to the whole biological community. Here, I only have several minor comments for the authors to consider.

1. The author demonstrated that Xenium has higher detection sensitivity than scFFPE-seq (suppl fig 2a-b) and Visium (fig 4f). I wonder whether the dropout genes in scFFPE-seq and Visium are random

or biologically biased. Please analyze whether gene dropout events are enriched in specific pathway categories beyond simple RNA abundance background.

2. Please include a figure comparing the gene dropout rates between scFFPE-seq and regular 10x 5' or 3' scRNA-seq on fresh samples. The authors do not have to generate new data, using a pool of previous 10x scRNA-seq datasets on fresh samples will be sufficient. This analysis will give potential users an expectation of gene count yield from the scFFPE-seq (depending on sequencing depth).

3. For the scFFPE-seq protocol, are there any cell types that cannot be captured? For example, are neutrophils preserved in dissociated FFPE tissues? I have not seen neutrophils in the t-SNE plot figure 2a.

4. Is there any established strategies to extract cell-cell communication information from Xenium data?

5. In the last part of the results (fig 7j), the author performed differential gene expression analysis of the six spots of triple-positive cells and all other Visium spots. How about comparing the six spots to other malignant spots and conducting a GSEA analysis? This might show the biological difference between different malignant cell states.

Reviewer #4:

Remarks to the Author:

In this manuscript, authors are exemplifying how new methods allow to analyze FFPE samples at unprecedented single cell and spatial resolution level. While it is clear that the technologies presented are allowing to deep insights of the intratumor heterogeneity, the advances in knowledge in breast tumor biology gained by the use of these technologies comes short in the manuscript. The manuscript reads more as a technical description of three methods rather than a study of breast cancer tumor microenvironment.

The main limit of the described work is the application of these techniques to only one tissue section, while the manuscript clearly demonstrates how one can use different techniques on the same FFPE sample to gain maximum insights of the single cell types present on a tissue section and their spatial distribution, it gives little new insights into the breast tumor biology, even though an attempt is made to study DCIS. The question remains if the few biological insights provided by the authors are generalizable and seen across different breast cancer subtypes or remain an isolated case. Validation of the findings is lacking!

The abstract is a bit long and may be divided into more succinct sections. My advice would be to decrease the lengths of the method section and would encourage the description of the novel findings. The case of low/high grade DCIS evolution towards IDC has been quite studied even spatially, the authors may need to describe more clearly what their analysis has allowed to demonstrate more clearly. It is however unsure if with the only tumor analyze here and the two different DCIS regions identified if the authors may be able to conclude and demonstrate novelty regarding the DCIS biology. The authors should cut short some of the sentences, paragraphs which to my opinion are dedicated to promotion of the methods presented here. For example, the section describing the first Xenium kit in the introduction is unnecessary. They should themselves expect their kits to evolve quite fast and this paragraph to be obsolete in a couple of years 7 months when scientists read their paper. Similarly, the use of wording like 'we introduce the novel ...' is not suited here I believe.

From the tSNE in Figure 2, it is complicated to follow the cell types annotations. In Figure 2A, it is difficult to see which clusters represent which cell types while in 2B which clusters are annotated, which are mixed, and which remain unannotated.

Besides, it seems they may be some cell types which are not seen in the dataset described by the

authors like NK cells and plasmablasts for example which have been clearly shown to be present in breast tumors analyzed by scRNA-seq data. Are these cell types 'not seen' by the scFFPE-seq? I think it would be valuable for the authors to look up a couple of landmark papers profiling breast tumors at the single cell level and try to harmonize their cell type annotation with these.

It is only in the discussion that the reader gets to understand how in Figure 2, the authors manage to identify DCIS cluster(s) from the scFFPE-seq already in figure 2 and that the cell type annotation of single cell is not totally agnostic of spatial location. When this understood, it actually sounds a bit circular that the authors claim to find the same cell type in Visium slides. If the spatial location is already used to annotate the single cell, no wonder the same cell types are found in the spatial transcriptomics analyses and at the expected location?

The authors identify two main DCIS regions #1 and #2, while around Fig6-7 they show different cell types composition for these two ROI, it is unclear to start with if the epithelial cells found in these two DCIS are showing phenotypic differences if yes which? We guess they should as they make-up different clusters? could these differences in epithelial cell phenotypes underlie the differences in microenvironment? Could this be assessed using algorithms inferring cell type interaction like cytoTalk or cellphoneDB?

The authors do not distinguish between normal and malignant epithelial cells. Are the cells from DCIS #1 #2 and all the invasive cells malignant and have genetic alterations? This should be looked upon and could further bring insights in differences between these epithelial cell types.

It is disappointing to see that authors are searching for the same cell types in the Xenium data than what has been identified in the scFFPE-seq, one could have hoped that the cell types mentioned above, plasmablasts, NK cells could be then identified if missing in the scFFPE data. In addition other rarer cell types which are harder to capture by single cells, adipocytes, granulocytes, neutrophils for example could have been seen? Despite the lower 'plexing' with that many cells 'measured' by Xenium, it would have been interesting to use the Xenium data more as a discovery set, unambiguously of the scFFPE data.

I believe that the method used to perform supervised labelling of the Xenium data from scFFPE-seq is missing. Sorry if I missed it somewhere.

I am surprised that adipocytes are not forming their own cluster in tSNE analysis despite their obvious distinct gene expression as pointed by the authors. Others performing sc-nuclei seq have isolated clusters of adipocytes, could that indicate inefficient clustering from the authors? and use of too low variable components for clustering?

While the authors replicate the Xenium they don't replicate the scFFPE, nor the Visium, why? It could be useful in their benchmarking efforts.

The authors claim that both scFFPE and Xenium are more sensitive than the 3' and 5' methods, first it is unclear on how this comparison was made, were the genes measured on the exact same cells to allow the plots to be drawn in supp fig 7, besides while we observe here a shift toward having scFFPE detecting more genes per cell it is unclear if this is significant as no statistics metrics are introduced. In Fig 4A the authors also claim that Xenium is more sensitive than scFFPE-seq, is it not expected that panel sequencing will be more sensitive than whole transcriptomics at identifying the genes in the panel? If the authors may agree with me here, to my mind all of figure 4 is an expected outcome. No statistics are introduced in Fig4 to compare sensitivity, etc..

'When we examined a region of atypical ductal hyperplasia (ADH) in this tissue, we were able to clearly localize key markers for eight different cell types (Fig. 4H-J; Supp. Fig. 9)'. Why do the authors mention this if no further comments, discussion relevance is mentioned after?

To me Fig 5 sound like a technical report, what does it brings apart from showing that Xenium allows to measure RNA and protein on the same section, does that belongs here, if the goal is to study breast cancer biology?

While the analysis of DCIS #1 and #2 ROI make sense, with the small number analyzed it is very difficult for the authors to substantiate their findings.

The manuscript lacks statistical assessments, as illustrated by the fact that their no statistical analysis paragraph in the Material and Method

Reviewer #5:

Remarks to the Author:

Summary: In this paper, the authors have demonstrated the integration of three commercially available genomic technologies (chromium, Visium, and Xenium) operating at different scales to capture insights into tumor progression and heterogeneity. The new technique introduced in this paper as Xenium In situ is a non-destructive platform achieving a subcellular spatial resolution that preserves the tissue for further staining/processing and is compatible with fresh frozen and FFPE tissues making it compatible with most of the specimens in the clinical pipeline. Xenium allows the simultaneous visualization of RNA and protein expression with high sensitivity and specificity. By combining the three platforms, they annotated the predominant cell types in the samples and derived high-resolution spatially resolved transcriptome data. The work is novel and provides a framework to uncover distinct tumor subtypes, sample tumor heterogeneity at the subcellular level and detect indicators of tumor invasion. However, the following comments should be addressed before publication.

- 1) Was there stage drift during multiple cycles of probe hybridization, imaging, and washing in the Xenium workflow? How did the authors detect and correct for it?
- 2) Was there any leaking of fluorescent light from one channel to another? What controls were done to ensure no spectral mixing? Please include these controls.
- 3) For the cell segmentation model, how did the authors account for overlapping/missed cells? Is there any estimate of the accuracy of the cell segmentation technique used by the authors?
- 4) Authors should show controls for the following:
 - a. Minimization of autofluorescence: It is mentioned that the background fluorescence was quenched using chemically using a proprietary quencher. Please show the autofluorescence images before and after quenching.
 - b. Effectiveness of washing of probes: authors need to include figures showing that unhybridized probes were successfully washed in each cycle and also show that the probe removal worked effectively across the 15 rounds of fluorescent probe hybridization.
 - c. Minimal off target effects: Include figures illustrating this
- 5) Can the Xenium system also comment on the abundance of a particular gene or transcript at a spatial location? Have authors attempted to estimate the mRNA expression profile and correlate it to the Xenium signal?

Point-by-point response below in blue text.

REVIEWER COMMENTS

Reviewer #2 (Remarks to the Author): Expert in spatial transcriptomics, computational genomics and epigenomics

The authors present a combination of technologies, including a new, commercially available Xenium technology, the existing commercial technologies Visium and Chromium. They apply these technologies to investigate an FFPE block of human breast cancer. While the technology is clearly impressive and the opportunities that arise with combining these three technologies, the study stays at a very superficial, exploratory level, mainly discussing advances - without comparing them to other current available methods, nor showing a concrete example of how these technologies can be used to advance medical research (as they claim in the discussion)

Overall, the paper reads more like an advertisement of the technologies, than a like a study that has a clear biological goal. This is of course fine, since technology development is crucial for advancing biology, however in this particular case, all the technologies are commercially available, which leaves the reader wondering what is the novelty of this.

We appreciate the reviewers' comments. We have taken measures to more explicitly highlight and expand on this work while at the same time amend the language used. We agree that technology development is critical, and while the technologies we present are now commercially available (they were not at the time of submission) there is yet to be a publication that describes them and the biological insights that they can provide hence we believe the work we present is novel. Please find our specific responses below.

- Can the authors point out how more clearly how the technologies they combined (each of them commercially available) can generate real biological insights?

The whole transcriptome single cell data allowed us to accurately type cells. Combining this with the Visium data provided spatial context but with limited resolution. Using the cell types derived from the single cell data and transferring those labels onto the Xenium data we were able to generate high resolution single cell spatial maps of the tissue. Once we have the high resolution spatial maps of the tissue we are able to explore the complex heterogeneity of the samples in this study.

An example of this is shown in Figure 4, where we highlight that one can derive whole transcriptome information using Visium from a region of interest that could only be identified with Xenium. The biological insight derived from such an analysis is substantial. We identified novel genes associated with the region of the human breast sample that was uniquely positive for three clinically relevant receptors (estrogen, progesterone, and HER2).

As part of our revisions we added another sample to the study and found that the integration of the technologies used in the study allowed us to see very fine biological detail that it was not possible to identify otherwise. In Figure 6, we identify a rare population of cells at the boundary of the DCIS region that express both myoepithelial and DCIS markers. We validate this with single cell data.

Both of these examples highlight how real biological insights can be made.

- the authors should also point out the limitations of the technology and where appropriate mention alternative technologies
- If the goal is to describe the technology, this should be done in more detail, also seriously pointing out limitations, competing technologies

We have added a sentence in the introduction stating that competing technologies exist. However it is beyond the scope of this manuscript to do a technology comparison as many of the technologies that are direct alternatives have extremely limited availability.

We point out limitations of all three technologies in the manuscript. This was partially the impetus for integrating three technologies, because each one individually has its own unique limitations. For example, the limitation of Visium is that it does not provide single cell resolution. A limitation of Chromium is that there is no spatial information. A limitation of Xenium is that it is not whole transcriptome, similar to all commercially available in situ technologies.

Nevertheless, we realize that there are other limitations that we failed to acknowledge explicitly. Cell segmentation in any in situ based technology is not a solved problem and limits the accurate annotation of certain cell types for which standard nuclei expansion is not good enough. For this reason, adipocytes are inconsistently annotated in Xenium datasets.

Another limitation we have added in the manuscript is that a large amount of input material is required (50-100 μm of the block) for scFFPE-seq. We acknowledge that not everyone can devote this much material to a single experiment, while still reserving material for spatial technology. However, the benefit to using scFFPE-seq over fresh sequencing of dissociated tumor cells is that there is faithful representation cell types in the section submitted for Xenium and Visium analyses.

- if the goal is to describe an application of the technology, the biological question should be more convincing - and exploratory analysis of just one breast FFPE block of one breast cancer sample can hardly be the way to learn something about breast cancer in general. There are some interesting leads in the paper that could be further explored, such as the detection of adipocytes
- if the goal is to present a data integration strategy, more focus should be on the data integration algorithms, which are currently not much explained in the main text.

Both goals are relevant for this study: application of the technology and data integration strategy. As this is not a large cohort study, we do not expect to make generalizable findings about human breast cancer. Due to the heterogeneity of both tissue and individual patient pathologies, our goal is not to generalize, but rather show the level of intricate detail in which one can explore their tissue of interest, and how molecular data with high gene plexy can be layered onto the pathology.

Specifically, we are interested in two concepts in this manuscript. We show the application of technology through the exploration of 2 human breast cancer examples which have a vast amount of cellular diversity within each sample. In a clinical setting,

the patient is not the “N”, but the single cells contained in the biopsy, and it is expected that diagnoses would be made based on the observations of one biopsy. That said, diagnostic concordance among pathologists could be improved. Breast cancer is heterogeneous, and our work demonstrates how more information can be gleaned from each biopsy, contributing to improved accuracy in diagnosis, prognosis, and hopefully early prevention.

The reviewer uses the word “exploratory” and we might infer that the reviewer is looking for hypothesis-driven work. We argue that there is merit in an experimental design which captures the biology of these individual samples at an unprecedented level of detail.

To address the comment that our data integration algorithms were not much explained in the main text, we have expanded our methods section in the revision, especially for the biology-driven Figures 4, 5 and 6, where data integration was key.

Overall, great technology, yet the fact that it is already commercially available makes it of course less novel, so I would think a more convincing biological question would be more appropriate to demonstrate the power of the combination of the proposed technologies.

The technologies we describe are very new to the field and as such there are no published reports that employ them (i.e. there are no Xenium or scFFPE-seq publications, only preprints), nor are there any that combine them to demonstrate what biological insights you can gain in the way we describe. scFFPEseq and Xenium have only very recently become commercially available. At the time of writing the original manuscript neither were available. However, whether commercially available or not, it does not make our study or the technologies we highlight less novel. While the technologies will provide high quality data, the tools to integrate and interpret the data and the know-how to explore such datasets, are still in their infancy.

Reviewer #3 (Remarks to the Author): Expert in spatial transcriptomics and computational genomics

The critical study by the Taylor group presents two novel technologies, in situ hybridization platform (Xenium In Situ) and the advanced single-cell gene expression workflow for FFPE samples (scFFPE-seq). Albeit lacking whole transcriptome profiling, Xenium provides higher gene detection sensitivity with subcellular spatial resolution. Integration of these technologies with the previous 10x Visium would be extremely powerful to explore deeper mechanisms of development and disease. For example, the authors apply them to serial breast cancer FFPE sections and find the molecular and cellular differences of tumor regions with and without invasion potential. These new platforms are exciting and promising to the whole biological community. Here, I only have several minor comments for the authors to consider.

1. The author demonstrated that Xenium has higher detection sensitivity than scFFPE-seq (suppl fig 2a-b) and Visium (fig 4f). I wonder whether the dropout genes in scFFPE-seq and Visium are random or biologically biased. Please analyze whether gene dropout events are enriched in specific pathway categories beyond simple RNA abundance background.

The lower sensitivity in the Visium assay is more systematic to the platform (per gene sensitivity) although we do believe we are evenly sampling from the true biological distribution of polyadenylated transcripts. While this could be partially resolved by

increasing the sequencing depth, because we have reached the plateau phase saturation the sequencing costs associated with picking up 1-2 more UMI counts per gene and a few more new genes with low counts would not contribute to gaining more complexity or biological insight.

Regarding sensitivity of Xenium...Xenium is, by and large, comparable to scFFPE-seq. We have removed the statement that "Xenium is 1.4x more sensitive than scFFPE-seq" since this will be experiment-dependent. Please also see our comments to Reviewer 4 on this subject.

2. Please include a figure comparing the gene dropout rates between scFFPE-seq and regular 10x 5' or 3' scRNA-seq on fresh samples. The authors do not have to generate new data, using a pool of previous 10x scRNA-seq datasets on fresh samples will be sufficient. This analysis will give potential users an expectation of gene count yield from the scFFPE-seq (depending on sequencing depth).

We agree that it is useful for people to know how the single cell technologies compare. We do expect a probe-based assay to have different sensitivities on a per cell basis than an RT-based assay, however, on average we see they are quite comparable (Supp Fig. 7). Since we know there is a lot of sample to sample variability, this comparison needs to be made on matched samples. To address the reviewers question we looked at the overlap between the genes with zero counts in each technology and have included a Venn diagram to highlight this in Supp Fig. 7C.

3. For the scFFPE-seq protocol, are there any cell types that cannot be captured? For example, are neutrophils preserved in dissociated FFPE tissues? I have not seen neutrophils in the t-SNE plot figure 2a.

We looked for neutrophils in the hBreast scFFPE-seq dataset but did not find a distinct cluster of cells expressing neutrophil markers (CSF3R, FPR1, FCGR3B, NAMPT, MND1A). However, we believe this is sample specific since we have seen them in other samples that we have run as part of our internal testing. For example, neutrophils can be found in mouse spleen FFPE tissue:

<https://www.10xgenomics.com/resources/datasets/10k-mouse-spleen-ffpe-tissue-dissociated-using-gentle-macs-dissociator-singleplex-sample-1-standard>

In this dataset, neutrophils are found in the cluster driven by the following markers: Ltfa, Camp, Itgb2l, Ngp, Cd177, and Abca13

Adipocytes are not a cell type that are easily detected with FFPE and or chop-fixed samples primarily because they float and do not pellet very well. Hence, they get lost during spins and washes. We have noted this in the revised manuscript and it is highlighted in Supp. Fig. 6.

4. Is there any established strategies to extract cell-cell communication information from Xenium data?

Xenium facilitates the detection of transcripts with subcellular resolution. Targeted approaches looking for ligand receptor pairs in adjacent cells would, of course, be possible if these genes were on a custom panel. This is not a question we address in this manuscript. We do expect that methods developed to extract cell-cell

communication for single-cell and in situ data will be applicable to Xenium. For example the StLearn developers already have a Visium specific vignette on cell-cell communication which could be modified for the Xenium data type.
<https://stlearn.readthedocs.io/en/latest/tutorials/stLearn-CCI.html>

5. In the last part of the results (fig 7j), the author performed differential gene expression analysis of the six spots of triple-positive cells and all other Visium spots. How about comparing the six spots to other malignant spots and conducting a GSEA analysis? This might show the biological difference between different malignant cell states.

We conducted this analysis and report our findings in Supplemental Figure 11.

Reviewer #4 (Remarks to the Author): Expert in spatial transcriptomics methods, imaging, pathology, and breast cancer tumour microenvironment

In this manuscript, authors are exemplifying how new methods allow to analyze FFPE samples at unprecedented single cell and spatial resolution level. While it is clear that the technologies presented are allowing to deep insights of the intratumor heterogeneity, the advances in knowledge in breast tumor biology gained by the use of these technologies comes short in the manuscript. The manuscript reads more as a technical description of three methods rather than a study of breast cancer tumor microenvironment.

We thank the reviewer for the comment. Our intention for this manuscript was not to perform a comprehensive study of the breast cancer tumor microenvironment across multiple patients and pathologies. Indeed, we initially limited ourselves to one sample (we have added an additional sample in our revisions) and do not expect to be able to make widely generalizable statements about novel aspects of breast cancer biology. Our goal is that this serves as more of a technical example of how experts in the field might take such approaches and apply to interesting biological questions and sample cohorts.

The main limit of the described work is the application of these techniques to only one tissue section, while the manuscript clearly demonstrates how one can use different techniques on the same FFPE sample to gain maximum insights of the single cell types present on a tissue section and their spatial distribution, it gives little new insights into the breast tumor biology, even though an attempt is made to study DCIS. The question remains if the few biological insights provided by the authors are generalizable and seen across different breast cancer subtypes or remain an isolated case. Validation of the findings is lacking!

To address this and expand the scope of the manuscript we have included an additional breast cancer sample run on Xenium that we explored in the same way (see Fig. 6). This figure further emphasizes the high granularity of cell types and the resolution at which we can view these cell types in situ.

The validation is provided by the agreement of orthogonal technologies. For example, in Figure 5, we identify a triple positive receptor region using Xenium, then we validate its presence using Visium and explore the whole transcriptome differential expression patterns further. In Figure 6, we identify a rare cell type in Xenium and validate the presence of these cells in the scFFPE-seq data.

We agree that there is an open question on generalizability but we would require a large cohort study in order to draw such conclusions and we hope that this study will be the impetus to drive such work in the field. As we describe in the manuscript, breast cancer is incredibly heterogeneous and with a sample size of only 2, we cannot make generalized conclusions.

We use the breast cancer samples to demonstrate what these novel technologies with high resolution and spatiality allow you to do, and highlight the huge potential for future discovery and translational research. While we identify some interesting insights in the heterogeneity of breast cancer that could be explored further, it is beyond the scope of the manuscript to do so.

The abstract is a bit long and may be divided into more succinct sections. My advice would be to decrease the lengths of the method section and would encourage the description of the novel findings.

We have revised the abstract to conform to the length requirements for Nature Communications and it is now significantly shorter. We have also moved technology benchmarking figures (which do not showcase biological insights, specifically) to supplemental. We've also highlighted our novel findings more, and included additional results from a different tissue section (Figure 6). We have not reduced the length of the methods section as we want to be fully transparent on our workflows and ensure readers have all the information they need to perform a similar study.

The case of low/high grade DCIS evolution towards IDC has been quite studied even spatially, the authors may need to describe more clearly what their analysis has allowed to demonstrate more clearly. It is however unsure if with the only tumor analyze here and the two different DCIS regions identified if the authors may be able to conclude and demonstrate novelty regarding the DCIS biology.

In one major spatial study, Risom et al., 2022 (<https://www.ncbi.nlm.nih.gov/pmc/articles/PMC8792442/>) observed differences in the tumor microenvironment among two different DCIS subtypes, characterized in part by myoepithelial gene expression. While Risom et al. is an elegant study, an important distinction is that they conducted bulk RNA-seq using laser capture microscopy in epithelial and stromal regions. Therefore, the distinct myoepithelial populations (KRT5+ and E-CAD+) they identified by antibody staining could not be further investigated at the whole transcriptome level to identify novel genes in those populations.

In contrast, we can derive whole transcriptome information for our two myoepithelial populations (KRT14+ and ACTA2+), predominantly via scFFPE-seq from serial sections, and also plausible with spot interpolation or deconvolution using Visium.

The authors should cut short some of the sentences, paragraphs which to my opinion are dedicated to promotion of the methods presented here. For example, the section describing the first Xenium kit in the introduction is unnecessary. They should themselves expect their kits to evolve quite fast and this paragraph to be obsolete in a couple of years 7 months when

scientists read their paper. Similarly, the use of wording like 'we introduce the novel ...' is not suited here I believe.

We have revised much of the text in the manuscript and improved the readability. We have also removed promotion of the methods and have included mention of other commercially available products. We also removed any forward looking statements as we agree with the reviewer that these would be out-of-date in the near future.

From the tSNE in Figure 2, it is complicated to follow the cell types annotations. In Figure 2A, it is difficult to see which clusters represent which cell types while in 2B which clusters are annotated, which are mixed, and which remain unannotated.

We have added labels to the legend in panel B. However note that Visium clusters could be mixtures of cell types depending on the region of the tissue a spot captures.

Besides, it seems they may be some cell types which are not seen in the dataset described by the authors like NK cells and plasmablasts for example which have been clearly shown to be present in breast tumors analyzed by scRNA-seq data. Are these cell types 'not seen' by the scFFPE-seq? I think it would be valuable for the authors to look up a couple of landmark papers profiling breast tumors at the single cell level and try to harmonize their cell type annotation with these.

These cells did not form a distinct cluster (see tSNE in Figure A below, not included in the manuscript) which is why they were not annotated as a distinct group, and then were not transferred to the Xenium data. We have looked at markers for NK cells and plasmablasts and do believe those cells are present in the scFFPE-seq data, just in small numbers. It is always challenging to define the exact clustering parameters (i.e. resolution, iterations, etc) to find all the expected cell types in independent clusters especially when cells are certainly undergoing transition from one type to another and might be intermediate cellular states.

The plasmablast marker CD38, is expressed in cells that cluster with B cells (see Figure C). This cluster of cells is MS4A1 negative (see Figure D), as expected for plasmablasts. NK cells are clustered with the CD8+ T cells (see Figure E) but are likely the cells circled in Figure E-H, marked by NKG7, KLRD1, and GNLY.

In our revisions, we have performed unsupervised labeling (see Figure 3K) of the Xenium data and can identify NK cells and plasmablasts, however we lose resolution in the tumor subtypes (e.g, DCIS #1 and #2 and proliferative tumor cells). We did find that in this clustering, NK cells are a subset of the CD8+ cells, and plasma cells are a subset of the B cells.

In Sample 2 (an additional sample we add for the revisions) we identify a variety of cell types including two different stromal populations, three myoepithelial, normal epithelial, a variety of immune cells and “boundary cells” which express both tumor and myoepithelial markers.

It is only in the discussion that the reader gets to understand how in Figure 2, the authors manage to identify DCIS cluster(s) from the scFFPE-seq already in figure 2 and that the cell type annotation of single cell is not totally agnostic of spatial location. When this understood, it actually sounds a bit circular that the authors claim to find the same cell type in Visium slides. If the spatial location is already used to annotate the single cell, no wonder the same cell types are found in the spatial transcriptomics analyses and at the expected location?

We used Visium and H&E data to aid in the annotation of our scFFPE-seq data. There are no universally accepted markers of DCIS cancer cells versus invasive cancer cells. Therefore, it makes sense that we would use spatial/histological data from Visium to assign the “DCIS” label to a particular single cell cluster.

Our analysis is intentionally iterative. We view this as an advantage since there was ample opportunity for validating the data across the three technologies, as well as the H&E and IF image registration. We view the built-in cross-platform redundancies as a strength.

For the revision, we have performed unsupervised labeling (see Figure 3K) of the Xenium data, which is independent of the scFFPE-seq annotation. While we did gain resolution in some immune cell populations, we lost resolution in the DCIS subtypes.

The authors identify two main DCIS regions #1 and #2, while around Fig6-7 they show different cell types composition for these two ROI, it is unclear to start with if the epithelial cells found in these two DCIS are showing phenotypic differences if yes which? We guess they should as they make-up different clusters? could these differences in epithelial cell phenotypes underlie the differences in microenvironment? Could this be assessed using algorithms inferring cell type interaction like cytoTalk or cellphoneDB?

Certainly, there are molecular/transcriptome differences between these two cell types (DCIS#1 and DCIS #2) since they cluster separately. As we discussed in the manuscript, the two DCIS regions look very similar to a pathologist, although it was noted that DCIS#2 had more proliferative cells and invasive lesions.

It would be speculation whether the epithelial cells dictate the microenvironment around them since cause/effect conclusions would require functional studies which are challenging to do in humans.

In a new Supplemental Figure 13, we provide an example whereby epithelial cells appear to be part of a normal breast duct. By all appearances, these cells look completely normal. However the molecular data show that many of the cells have converted to a tumor cell type.

Targeted approaches looking for interactions between cells would be possible if the relevant genes of interest (e.g., ligand-receptor pairs) were on a custom panel. This is not a question we address in this manuscript. We do expect that methods developed to extract cell-cell communication for single-cell and in situ data will be applicable to Xenium. For example the StLearn developers already have a Visium specific vignette on cell-cell communication which could be modified for the Xenium data type. <https://stlearn.readthedocs.io/en/latest/tutorials/stLearn-CCI.html>

The authors do not distinguish between normal and malignant epithelial cells. Are the cells from DCIS # 1 #2 and all the invasive cells malignant and have genetic alterations? This should be looked upon and could further bring insights in differences between these epithelial cell types.

We feature a new human breast cancer section (Sample #2) in our revision where normal duct cells are clearly present. Normal epithelial cells have a gene signature distinct from malignant cells, expressing *ESR1* and *SERPINA3*. In contrast, tumor epithelia express *ERBB2* and *ABCC11*. See Figure 6 and Supplemental Figure 12.

In the original biological section (Sample #1), all the epithelial cells (EPCAM+) in DCIS and invasive regions were malignant, as confirmed by a pathologist.

We are unable to detect genetic alterations given that genomic DNA was not collected for our experiments.

It is disappointing to see that authors are searching from the same cell types in the xenium data than what has been identified in the scFFPE-seq, one could have hoped that the cell types mention above, plasmablasts, NK cells could be then identified if missing in the scFFPE data. In addition other rarer cell types which are harder to capture by single cells, adipocytes, granulocytes, neutrophils for example could had been seen? Despite the lower 'plexing' with that many cells 'measured' by xenium, it would had been interesting to use the Xenium data more as a discovery set, unambiguously of the scFFPE data.

We are not using Xenium as a discovery platform here, since it is a targeted assay with well-defined panels which are designed to interrogate specific tissues and expected cell types within. In our revisions we perform unsupervised clustering of the Xenium data and do identify additional cell types that we were not able to distinguish using the supervised labelling from the scFFPE-seq data (see Figure 3K).

We were not searching for the same cell types in Xenium. In the original version of the manuscript (and currently Figure 5) we characterized a cell type expressing PGR that was in very small numbers within the section. We argued that Xenium identified this rare cell type when scFFPE-seq failed to do so. In the new Figure 6, Xenium identifies a rare cell type that co-expresses both myoepithelial and tumor markers.

I believe that the method used to performed supervised labelling of the Xenium data from scFFPE-seq is missing. Sorry if I missed it somewhere.

There is a section in the methods section entitled, "Supervised labeling & label transfer" which describes how the supervised label transfer was performed.

I am surprised that adipocytes are not forming their own cluster in tSNE analysis despite their obvious distinct gene expression as pointed by the authors. Others performing sc-nuclei seq have isolated clusters of adipocytes, could that indicate inefficient clustering from the authors? and use of to low variable components for clustering?

Adipocytes are not a cell type that are easily detected with FFPE and or chop-fixed samples primarily because they float and do not pellet very well. Hence, they get lost during spins and washes. We believe this is a sample prep problem and not a clustering problem.

While the author replicate the Xenium they don't replicate the scFFPE, nor the Visium, why? It could be useful in their benchmarking efforts.

The reason for this (as described in methods) is that scFFPE-seq requires a large quantity of material (50-100 μm of FFPE curls off the microtome) for each replicate. It was essential that we reserved the block for Visium and Xenium experiments, as well as initial quality controls. Our scFFPE-seq data is 4 technical replicates that are aggregated together. In the figure below, we have broken out each technical replicate to demonstrate the almost perfect correlation (mean expression between genes is 0.999).

The authors claim that both scFFPE and Xenium are more sensitive than the 3' and 5' methods, first it is unclear on how this comparison were made, were the genes measured on the exact same cells to allow the plots to be draw in supp fig 7, beside while we observe here a shift toward having scFFPE detecting more genes per cell it is unclear if this is significant as no statistics metrics are introduced.

Given the current capabilities of these technologies it is impossible to conduct all three assays on the same exact cells. scFFPE-seq and Xenium are conducted on serial sections, and therefore we cannot do sensitivity comparisons on the same cells. Rather, we did a bulk comparison, normalized to the number of cells and read depth.

The sensitivity measurements for both scFFPE-seq and Xenium are unique for each gene because they are probe based assays. Thus, a good metric to describe the effective sensitivity is an average or median gene sensitivity (a distribution of sensitivity estimates). We added this to the methods under "Benchmarking sensitivity".

In Fig 4A the authors also claim that Xenium is more sensitive than scFFPE-seq, is it not expected that panel sequencing will be more sensitive than whole transcriptomics at identifying the genes in the panel? If the authors may agree with me here, to my mind all the figure 4 is an expected outcome. No statistics are introduced in Fig4 to compare sensitivity, etc..

Since scFFPE-seq and Xenium have fundamentally different chemistries with different limitations we did not expect that the targeted approach would necessarily be more sensitive than whole transcriptome analysis. There are many components in both the Xenium and scFFPE-seq assays which contribute to the observed gene counts: the

cells' original RNA copy number, sample preparation, RNA quality, gene sequence content, probe binding kinetics, polymerase kinetics etc.

Furthermore, panel size is a major factor, as is the number of probes per gene. If you run 10 genes with 16 probes for each you would get higher sensitivity per gene than if you ran a panel with 500 genes and 4 probes for each. The same is true for all optical read out technologies. There are physical and budgeting limitations that occur to balance the targeted gene plexy with sensitivity. Some researchers will prioritize plexy, others will prioritize sensitivity, and others will prioritize multi-modality outputs (e.g., protein). It completely depends on the scientific question or application.

All of these factors mentioned above contribute to the observation of each gene's counts in each assay. It follows that the mean count for an individual gene is an estimate of the objective RNA count convolved with sample quality and assay sensitivity.

Both scFFPE-seq and Xenium were performed on serial sections from the same biopsy, thus controlling for sample quality. By taking the ratio in mean counts for each gene between assays, we normalize out the contribution of objective RNA count and sample quality. Thus, the ratio in counts for a specific gene is an estimate of the assay sensitivity. By evaluating the distribution of sensitivity estimates across genes we can get an idea about the relative sensitivities of the two assays by averaging out the gene specific variability.

We acknowledge that such an analysis is still imperfect because there are unique components of sample quality that impact the assays in different ways. For example, tissue autofluorescence impacts Xenium but not scFFPE-seq. A more refined estimate of the relative sensitivities of the two assays would require many samples spanning different tissue types and qualities, which is beyond the scope of this manuscript.

We have added more detail to the methods regarding Xenium algorithms including cell segmentation, decoding, quality scores, controls, etc. also adding a whole new differential gene expression analysis section. We have also removed the sentence that Xenium is 1.4x more sensitive and instead say that the two platforms are comparable because this will vary across experiments, panel design, and plexy.

'When we examined a region of atypical ductal hyperplasia (ADH) in this tissue, we were able to clearly localize key markers for eight different cell types (Fig. 4H-J; Supp. Fig. 9)'. Why the authors mention this if no further comments, discussion relevance is mentioned after?

We have since moved this figure to supplemental as it was an exploration on the sensitivity, specificity, and resolution of the technologies, and not biology focused.

To me Fig 5 sound like a technical report, what does it brings apart from showing that Xenium allows to measure RNA and protein on the same section, does that belongs here, if the goal is to study breast cancer biology?

Yes, it is important to have RNA and protein on the same section as protein detection can identify cell types that are nearly impossible to characterize as RNA expression levels are typically extremely low. Here, it contributed to the biology story by confirming

that the block is HER2 positive. However, in order to allow for more biology focused figures in the main text we have moved this figure to Supplemental.

While the analysis of DCIS #1 and #2 ROI make sense, with the small number analyzed it is very difficult for the authors to substantiate their findings.

We include another sample in this revision, but would like to reiterate that we are not trying to find generalizable trends but to demonstrate what can be discovered and what is being missed with the current state of the art. Our goal is to highlight the potential for those discoveries to be impactful in diagnostics/the clinic down the road.

The manuscript lacks statistical assessments, as illustrated by the fact that their no statistical analysis paragraph in the Material and Method

We have added more analysis details throughout the manuscript and have added a section in the methods on how p-values were calculated for the differential gene expression analysis.

Reviewer #5 (Remarks to the Author): Expert in spatial transcriptomics methods, imaging, pathology, breast cancer tumour microenvironment, and artificial intelligence

Summary: In this paper, the authors have demonstrated the integration of three commercially available genomic technologies (chromium, Visium, and Xenium) operating at different scales to capture insights into tumor progression and heterogeneity. The new technique introduced in this paper as Xenium In situ is a non-destructive platform achieving a subcellular spatial resolution that preserves the tissue for further staining/processing and is compatible with fresh frozen and FFPE tissues making it compatible with most of the specimens in the clinical pipeline. Xenium allows the simultaneous visualization of RNA and protein expression with high sensitivity and specificity. By combining the three platforms, they annotated the predominant cell types in the samples and derived high-resolution spatially resolved transcriptome data. The work is novel and provides a framework to uncover distinct tumor subtypes, sample tumor heterogeneity at the subcellular level and detect indicators of tumor invasion. However, the following comments should be addressed before publication.

1) Was there stage drift during multiple cycles of probe hybridization, imaging, and washing in the Xenium workflow? How did the authors detect and correct for it?

We perform fiducial registration each cycle that corrects for stage drift.

2) Was there any leaking of fluorescent light from one channel to another? What controls were done to ensure no spectral mixing? Please include these controls.

Filters have been designed to minimize crosstalk between channels. That said, there is some residual spectral cross-talk between imaging channels. We mitigate this effect in two ways. Let's say we have two channels X and Y and a rolling circle product (RCP) is expected to be detected in channel X in a particular cycle. Due to spectral cross-talk (which we expect to be minimal), we also see the same RCP in channel Y, but dimmer. Because the decoding algorithm utilizes intensity information, the dim RCP in channel Y is accounted for by our probabilistic intensity-based decoding algorithm. Secondly, real RCPs must be detected in a pre-specified pattern (the codebook) and spectral cross-talk will produce a lower quality-score codeword assignment as compared to a "real" RCP.

We added more information about Xenium controls and decoding algorithms in the method's section.

3) For the cell segmentation model, how did the authors account for overlapping/missed cells? Is there any estimate of the accuracy of the cell segmentation technique used by the authors?

To address the first question, we do detailed comparisons to hand drawn ground truth as part of the training and benchmarking process. For nuclei with large overlaps (intersection-over-union ≥ 0.05), the nucleus with better focus is selected as the nucleus to use and others are removed; for nuclei with small region of overlaps (intersection-over-union < 0.05), all nuclei are kept and the overlapping region is assigned to a (considered random) nucleus.

To address the second question, we have looked at % transcripts assigned to cells, although this is more a metric for sensitivity rather than specificity. More details have been added on cell segmentation in the methods section.

4) Authors should show controls for the following:

a. Minimization of autofluorescence: It is mentioned that the background fluorescence was quenched using chemically using a proprietary quencher. Please show the autofluorescence images before and after quenching.

We do not take images before quenching since it is performed prior to decoding. We also cannot afford to run every experiment plus and minus quencher so we cannot include this in the manuscript for the specific samples used. Please see below, for example, images of human breast cancer plus and minus quenching on serial sections different from the block featured in the manuscript. Scale bar = 1 mm.

b. Effectiveness of washing of probes: authors need to include figures showing that unhybridized probes were successfully washed in each cycle and also show that the probe removal worked effectively across the 15 rounds of fluorescent probe hybridization.

We have acquired an image after cycle 15 and have shown that there is no signal from RCPs, only autofluorescence (see below; Scale bar = 0.5 mm). The fact it looks clean after 15 cycles implies that it was clean across all cycles.

Our internal computational pipeline has a metric for stripping efficiency (see figure below). That is, how often do you see the same RCP of the same color in back-to-back cycles. We would prefer not to add this figure to the main manuscript because it is not part of our customer-facing web summary, and it detracts from the main story.

C.

Minimal off target effects: Include figures illustrating this

Please see Supplemental Figure 2B. This figure shows 1) probe controls to assess non-specific binding to RNA, 2) decoding controls to assess misassigned genes, and 3) genomic DNA (gDNA) controls to ensure the signal is from RNA.

5) Can the Xenium system also comment on the abundance of a particular gene or transcript at a spatial location? Have authors attempted to estimate the mRNA expression profile and correlate it to the Xenium signal?

Like other single-cell technologies (e.g., scRNA-Seq, scFFPE-seq), Xenium in situ data analysis does not detect every transcript of a given gene present in a cell, which leads to stochastic variation in the observed number of transcripts per cell. However we observe a very high correlation between Xenium and scFFPE-seq expression profiles for genes expressed at levels spanning more than three orders of magnitudes (see Supplemental Figure 7D), which indicates that Xenium is as quantitative as single cell technologies for measuring gene expression levels.

Reviewers' Comments:

Reviewer #2:

Remarks to the Author:

The authors have significantly improved the manuscript and specifically now provide more details about the methods. The only part where reproducible methods are still missing is the part about Cell segmentation, where authors used a "custom neural network for nucleus segmentation". Ideally, the code, or even better the trained network would be made available with the paper. At the very least they should describe the neural network in more detail.

Reviewer #3:

Remarks to the Author:

The authors have addressed my questions. I have no further questions and believe that this work presents very useful spatial genomics technologies.

Reviewer #4:

Remarks to the Author:

Major comments

1. In one sample, we identify three molecularly distinct tumor subtypes, enabling us to define cell neighborhoods and biomarkers in the progression towards invasive carcinoma. This sentence in the abstract may be a bit difficult to understand. In this sentence, I would potentially if there is space highlight that two of them were ductal carcinoma in situ and one invasive. This sentence may also be a bit reductive to a certain point of view as there are potentially much more tumor ecosystems / tumor subtypes found in the sample analyzed than three?"
2. In Figure 2B and 2A, while the authors have clarified that the Visium data helps to annotate the DCIS #1 and #2 in the scFFPE, it is unclear what technique, method they employ to make sure that the DCIS that are first morphologically identified in the Visium which allows to clearly get these DCIS transcriptome, are then transferred to the scFFPE data. Is it by transcriptional similarity? Is it the label transfer section in the method 'Supervised labeling & label transfer'? If yes it should be stated or is it just through correlation/similarity of gene expression profiles?
3. While I acknowledge that pathologists can distinguish between normal and malignant epithelial cells morphologically in HE stains, it gets more uncertain when looking at DCIS, would it be relevant to use an algorithm like inferCNV or copyKAT to investigate whether DCIS #1 and #2 seem to have copy number alterations which could help the distinction between normal and malignant epithelial cells and also maybe explain the differences in transcriptome and phenotype?"
4. Would be valuable to identify a couple of markers which allow to distinguish between Macrophages 1 and 2 for researchers in the future.
5. As pinpointed previously it is unclear how the different technologies treat different cell types. While the authors have clarified the matter around the adipocytes, it would be valuable to see proportions of the different cell types according to different technologies. It would be valuable for researchers to know what to expect when using each technology. They could, in the future, adapt the technology according to the cell type they want to study most if some technologies are better at profiling different types than others. Of course, for Xenium, it will highly depend on the panel used.
6. In the future, researchers may try to cluster their Xenium data. Should they expect less resolution in cell types than when performing clustering with scFFPE with only the genes they use for Xenium? I

understand this will depend on clustering parameters, etc., but here authors have supposedly done the same and Xenium seems less resolute?

7. Authors seem to indicate that there would be in Xenium some 'doublets' according to proximity. Could these be identified through the expression of mixed marker genes or the number of genes identified or the number of reads in a 'segment'? Could authors provide a way to identify them in the future, for future researchers having only Xenium data which is likely to happen?

8. Intriguingly, there were two regions where the histology appeared to have normal duct morphology, but the molecular data revealed tumor cell markers (Supp. Fig. 13). This suggests that the Xenium data can provide insight as to whether a duct will progress towards a carcinoma prior to morphological changes detectable by a pathologist. This sentence is highly speculative, without data on genetic aberration it is impossible to conclude. Of course, here with only panel inference of copy number is impossible.

Minor comments

- I believe some figures are mislabeled in the revised manuscript fig 3J \diamond Fig 3M in page 8. Revised page 8 and how figure 3 is mentioned throughout the text

- Legend in Fig 6a is missing. Which makes it for example difficult to follow if the CTLA4+ cells have a different color or are just quickly added to the Figure

Reviewer #5:

Remarks to the Author:

All reviewer comments have been addressed, the manuscript can be accepted for publication.

REVIEWERS' COMMENTS

Reviewer #2 (Remarks to the Author):

The authors have significantly improved the manuscript and specifically now provide more details about the methods. The only part where reproducible methods are still missing is the part about Cell segmentation, where authors used a "custom neural network for nucleus segmentation". Ideally, the code, or even better the trained network would be made available with the paper. At the very least they should describe the neural network in more detail.

We have rewritten the Cell Segmentation section in the methods in order to provide more details. We are unable to make the full code available because this is proprietary. However, we have added three references to the Methods section which provide more context and foundation for the custom neural network code.

We have also added two links to documentation on the Xenium Ranger software (10x Genomics). For users who wish to avoid using our proprietary algorithms, the **import-segmentation** pipeline (<https://www.10xgenomics.com/support/software/xenium-ranger/analysis/running-pipelines/XR-import-segmentation>) allows researchers to import their own segmentation results generated by third party tools (e.g., Baysor). Users can also change two segmentation parameters (nucleus expansion distance, minimum DAPI intensity) with the **resegment** pipeline (<https://www.10xgenomics.com/support/software/xenium-ranger/analysis/running-pipelines/XR-resegment>).

Reviewer #3 (Remarks to the Author):

The authors have addressed my questions. I have no further questions and believe that this work presents very useful spatial genomics technologies.

We thank the reviewer for their earlier comments which improved the manuscript.

Reviewer #4 (Remarks to the Author):

Major comments

1. *In one sample, we identify three molecularly distinct tumor subtypes, enabling us to define cell neighborhoods and biomarkers in the progression towards invasive carcinoma.*

This sentence in the abstract may be a bit difficult to understand. In this sentence, I would potentially if there is space highlight that two of them were ductal carcinoma in situ and one invasive. This sentence may also be a bit reductive to a certain point of view as there are potentially much more tumor ecosystems / tumor subtypes found in the sample analyzed than three?"

We have reworded the abstract (including this sentence) to meet the word count and to add clarity. Since there was not space to go into detail on the samples we have modified this section to focus more on what our approach allowed us to learn.

This section of the abstract now reads:

This integrative approach allowed us to explore molecular differences that exist between distinct tumor regions, and to identify biomarkers involved in the progression towards invasive

carcinoma. Further, we study cell neighborhoods and identify rare boundary cells that sit at the critical myoepithelial border confining the spread of malignant cells.

2. In Figure 2B and 2A, while the authors have clarified that the Visium data helps to annotate the DCIS #1 and #2 in the scFFPE, it is unclear what technique, method they employ to make sure that the DCIS that are first morphologically identified in the Visium which allows to clearly get these DCIS transcriptome, are then transferred to the scFFPE data. Is it by transcriptional similarity? Is it the label transfer section in the method 'Supervised labeling & label transfer'? If yes it should be stated or is it just through correlation/similarity of gene expression profiles?

We did this through transcriptional similarity and noted this in the text.

For example, DCIS #1 exclusively expresses *TTC3* and DCIS #2 exclusively expresses *CPB1*. These genes also define the DCIS #1 and DCIS #2 clusters in the single-cell data.

(A) Dimension reduction of the scFFPE-seq data yielded a t-SNE projection with 17 unsupervised clusters. DCIS #1 and #2 are distinct clusters. (B, C) t-SNE projection of *TTC3* which marks DCIS #1 and *CPB1* which marks DCIS #2. (D) H&E staining conducted pre-CytAssist is shown for reference alongside the spatial distribution of clusters in (E, F). The Visium data elucidated the spatial location of two molecularly distinct DCIS, marked here by (E) *TTC3* and (F) *CPB1*, respectively.

We did not have a methods section devoted to this specifically because Visium annotation was not computationally integrated in the same way that Xenium was annotated via supervised label transfer from scFFPE-seq.

3. While I acknowledge that pathologists can distinguish between normal and malignant epithelial cells morphologically in HE stains, it gets more uncertain when looking at DCIS, would it be relevant to use an algorithm like inferCNV or copyKAT to investigate whether DCIS #1 and #2 seem to have copy number alterations which could help the distinction between normal and malignant epithelial cells and also maybe explain the differences in transcriptome and phenotype?"

We agree with the reviewer in that the classification of different subtypes of DCIS is challenging for a pathologist. The point we are making is exactly that, on initial inspection of the tissue the pathologist was able to classify normal, invasive and DCIS regions. However, when we shared the molecular data with the pathologist from the Xenium run, the pathologist noted that there were some morphological differences between the DCIS regions, as described in the Results section with header, "Exploration of a breast carcinoma sample with three distinct tumor subtypes reveals heterogeneity in myoepithelial, immune, and invasive cell populations."

Exploring copy number alterations is an interesting avenue to pursue, but it is beyond the scope of this manuscript.

4. Would be valuable to identify a couple of markers which allow to distinguish between Macrophages 1 and 2 for researchers in the future.

Please find below a heatmap of differentially expressed genes between macrophages #1 and #2. We have added two sentences to the Figure 2 legend identifying some key markers that distinguish these two types of macrophages. Given that Macrophages #2 express CD163, it is possible that these are the M2 macrophages, or "alternatively activated".

5. As pinpointed previously it is unclear how the different technologies treat different cell types. While the authors have clarified the matter around the adipocytes, it would be valuable to see proportions of the different cell types according to different technologies. It would be valuable for researchers to know what to expect when using each technology. They could, in the future, adapt the technology according to the cell type they want to study most if some technologies are better at profiling different types than others. Of course, for Xenium, it will highly depend on the panel used.

We agree with the reviewer that certain technologies will be more appropriate for targeting specific cell types. However, it is beyond the scope of this manuscript to be able to make a conclusive statement on this subject given that the sample size was small.

We also agree that the capture of different cell types will be dependent on the Xenium panel, and it will also depend on the tissue type and biopsy region.

We provide cell proportions below, however, there is likely sampling bias because 5' GEX and 3' GEX were performed on dissociated tumor cells from the same patient, but might not be the same region of the tissue biopsy used in Xenium, Visium, and scFFPE-seq.

6. In the future, researchers may try to cluster their Xenium data. Should they expect less resolution in cell types than when performing clustering with scFFPE with only the genes they use for Xenium? I understand this will depend on clustering parameters, etc., but here authors have supposedly done the same and Xenium seems less resolute?

We clustered the Xenium data in Figures 3H, 3K and 6A. In Figure 3G, we down-selected the scFFPE-seq data to the 313 genes on the Xenium panel.

The short answer is, yes, using fewer genes (1.8% of the whole transcriptome) will certainly be less resolute when clustering the scFFPE-seq data. An interesting question beyond the scope of this manuscript would be to understand how many genes and which genes you would need to be at, say, 90% resolving power of scFFPE-seq. One could generate confusion matrices between the clustering of whole transcriptome and downsampled data, tested with different downsampling rates or a different composition of genes.

In the manuscript, in a similar analysis, we performed unsupervised clustering of the Xenium data (without integrating scFFPE-seq) and we lost resolution in the DCIS subtypes. An oncology-specific panel or custom panel would likely refine the tumor subtypes as well as other cell types. In lieu of using a panel tailored to our specific tissue, we integrated the data with scFFPE-seq to improve annotation/resolution of cell types.

7. Authors seem to indicate that there would be in Xenium some 'doublets' according to proximity. Could these be identified through the expression of mixed marker genes or the number of genes identified or the number of reads in a 'segment'? Could authors provide a way to identify them in the future, for future researchers having only Xenium data which is likely to happen?

Since the Xenium technology is imaging based and requires segmentation of cells in order to assign transcripts to cells, it is possible to have errors in segmentation that result in transcripts from more than one cell being assigned to a single cell. Currently, the best way to assess this is by manual inspection of the cells of concern, using knowledge of local cell types to determine if there are any incorrectly segmented cells. In the future Xenium, and other platforms like it, will move towards cell segmentation that is based on membrane boundary stains which will minimize segmentation errors like those that can arise from nuclear expansion. Also, 3D segmentation would be advantageous, but how to approach this is an open question in the field.

8. Intriguingly, there were two regions where the histology appeared to have normal duct morphology, but the molecular data revealed tumor cell markers (Supp. Fig. 13). This suggests that the Xenium data can provide insight as to whether a duct will progress towards a carcinoma prior to morphological changes detectable by a pathologist.

This sentence is highly speculative, without data on genetic aberration it is impossible to conclude. Of course, here with only panel inference of copy number is impossible.

The point we want to make here is that RNA expression alone can identify a normal or cancerous cell, before morphological distinction can be made. These distinctions are made routinely in the single cell field where tumor cells cluster away from normal epithelial cells. In Supplemental Figure 13 we show a region that is morphologically normal (as assessed by pathologist), yet the molecular profile of some of these cells are cancerous. Information about genetic aberrations or copy number are not required here since transcript expression is a manifestation of underlying genetic changes.

Minor comments

- I believe some figures are mislabeled in the revised manuscript fig 3J Fig 3M in page 8. Revised page 8 and how figure 3 is mentioned throughout the text

Thank you for bringing this to our attention. We have fixed the incorrect call-outs in the Figure 3 text body of the Results section.

- Legend in Fig 6a is missing. Which makes it for example difficult to follow if the CTLA4+ cells have a different color or are just quickly added to the Figure

We labeled some markers in the UMAP like *CTLA4*, *CD83*, and *TOP2A* because they were found as a result of subclustering the initial UMAP. We decided not to assign vividly distinct colors to these clusters in order to reserve a diverse color palette for the cell types we focused on in the remainder of the figure, as well as Supplemental Fig. 13. Since immune cells were not the focus of these figures, we did not vary the colors extensively. We have drawn a circle around the *CTLA4*⁺ cells and darkened the color of the cluster for clarity and added a legend to Fig. 6A as requested.

Reviewer #5 (Remarks to the Author):

All reviewer comments have been addressed, the manuscript can be accepted for publication.

We thank the reviewer for their earlier comments which improved the manuscript.